# LVLM-Count: Enhancing the Counting Ability of Large Vision-Language Models

**Muhammad Fetrat Qharabagh**                     *m2fetrat@uwaterloo.ca*
*Cheriton School of Computer Science*
*University of Waterloo*

**Mohammadreza Ghofrani**
*Independent Researcher*

**Kimon Fountoulakis**                     *kimon.fountoulakis@uwaterloo.ca*
*Cheriton School of Computer Science*
*University of Waterloo*

**Reviewed on OpenReview:** *https://openreview.net/forum?id=G1i9MUQj63*

## Abstract

Counting is a fundamental operation for various real-world visual tasks, requiring both object recognition and robust counting capabilities. Despite their advanced visual perception, large vision-language models (LVLMs) are known to struggle with counting tasks. In this work, we evaluate the performance of several LVLMs on visual counting tasks across multiple counting and vision datasets. We observe that while their performance may be less prone to error for small numbers of objects, they exhibit significant weaknesses as the number of objects increases. To alleviate this issue, we propose a simple yet effective baseline method that enhances LVLMs' counting ability for large numbers of objects using a divide-and-conquer approach. Our method decomposes counting problems into sub-tasks. Moreover, it incorporates a mechanism to prevent objects from being split during division, which could otherwise lead to repetitive counting—a common issue in a naive divide-and-conquer implementation. We demonstrate the effectiveness of this approach across various datasets and benchmarks, establishing it as a valuable reference for evaluating future solutions.

## 1 Introduction

Counting is a key cognitive task with broad applications in industry, healthcare, and environmental monitoring (De Almeida et al., 2015; Guerrero-Gómez-Olmedo et al., 2015; Paul Cohen et al., 2017; Lempitsky & Zisserman, 2010). It improves manufacturing, inventory, and quality control, ensures safety in medical settings, and helps manage resources in environmental efforts (Wang & Wang, 2011; Zen et al., 2012; Arteta et al., 2016). These applications often require distinguishing between objects of the same class with subtle variations, as well as recognizing complex concepts. Models trained solely on counting datasets struggle to generalize to such scenarios due to the limited availability of annotated data for fine-grained counting. Recent advancements in large vision-language models (LVLMs), such as GPT-4o (Achiam et al., 2023), combined with the massive scale of web-scraped training data, have enabled unprecedented zero-shot recognition capabilities, making them a promising candidate for handling complex and fine-grained counting tasks. However, evaluations of LVLMs have also revealed notable weaknesses in their numerical reasoning (Yin et al., 2023; Xu et al., 2024; Yang et al., 2023).

In this work, we focus specifically on the visual counting, one of the most fundamental aspects of numerical reasoning. We observe that although LVLMs perform well in counting small numbers of objects—typically fewer than 20—their accuracy deteriorates with larger quantities. Inspired by prior work on the rapid and

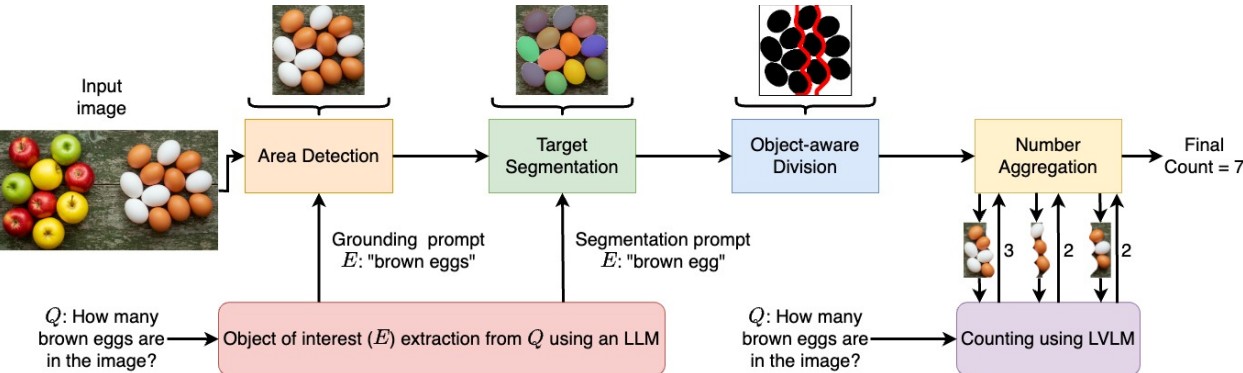

Figure 1: Illustration of our proposed pipeline. First, an expression ($E$) describing the area of interest is extracted from the prompted question ($Q$), such as "brown eggs". The expression is extracted using a large language model (LLM) which is the same as LVLM in our work. Then, $E$ and the image are provided as input to a grounding model, such as the one by Liu et al. (2023) to detect the area of interest. Second, any objects corresponding to $E$ are segmented. Third, in the object-aware division step, we use the segmentation masks to divide the detected area of interest without cutting through the objects of interest. Finally, the number of objects of interest in each sub-image is computed using an LVLM, and the results are aggregated.

accurate estimation of small quantities by Chattopadhyay et al. (2017) we propose a simple yet effective baseline method to alleviate this issue. Leveraging a divide-and-conquer approach, we engineer a simple pipeline that divides an image into carefully cut sub-images, and prompts the LVLM to count the objects of interest in each sub-image. The counts from the sub-images are then aggregated to make the final prediction. A key feature of our pipeline is a mechanism that prevents objects of interest from being split by the dividing lines, which could otherwise lead to double-counting. To have this mechanism two existing pre-trained detection and segmentation models are employed in the pipeline. The workflow is illustrated in Figure 1.

Initially, in our pipeline, the category name of the object of interest is extracted from the input question using an LLM (We use the LVLM as an LLM for this step). The area containing the objects of interest is detected in the image by a grounding model, such as Liu et al. (2023), and then cropped. The cropping step removes irrelevant context from the image. Secondly, using an object detection model by Liu et al. (2023), and a segmentation model by Kirillov et al. (2023), the segmentation masks of the objects of interest are created. Thirdly, we use a mechanism that divides the image into multiple sub-images without cutting through the objects of interest. We call this mechanism object-aware division. The division positions are determined automatically using an unsupervised and non-parametric method based on object masks. Then we treat the object-aware division as a path-finding problem, avoiding the segmented objects as obstacles. A black-white image is built by converting all the masks into black and the rest of the image into white pixels. The binary image is converted into a graph where only white pixels are connected as nodes. Using the $A^*$ algorithm (Russell & Norvig, 2016), a path is found from one end to the other end of the image, ensuring objects remain intact. Finally, using an LVLM as a counting tool, the objects of interest in the sub-images are counted and aggregated. Our contributions are summarized below:

1. We evaluate the counting performance of several recent Large Vision-Language Models (LVLMs) on multiple counting and vision datasets. We propose a simple yet effective baseline method, LVLM-Count, which enhances the counting performance of LVLMs without requiring additional training. Similar to standard counting with LVLMs, our method is a prompt-based approach that retains their zero-shot capabilities while addressing their difficulties in handling large numbers. Through experiments, we demonstrate that LVLM-Count improves LVLMs' counting performance across the evaluated datasets.

2. We propose a solution for object-aware division. Accurate division is crucial, as parts of cut objects can lead to over-counting (see Figure 2a). The proposed solution divides images without cutting through objects of interest specified by an arbitrary prompt.

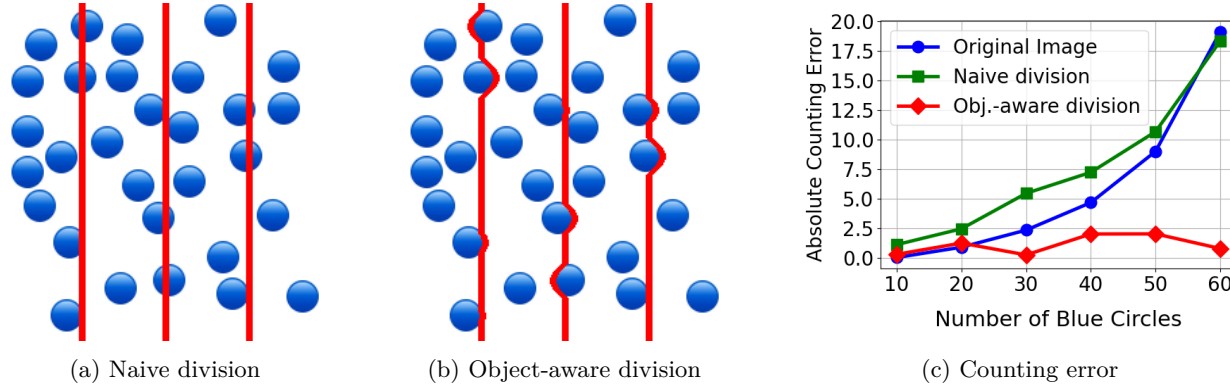

Figure 2: Comparison of the naive and the object-aware division. The objects of interest are the circles. In Figure 2a, we illustrate a naive division of the input image, which is divided into equally sized sub-images with straight lines. In Figure 2b, we illustrate the object-aware division, which avoids cutting through circles. In Figure 2c, we illustrate the counting error of GPT-4o for images with randomly positioned circles. The absolute counting error is the absolute difference between the ground truth and the number predicted by GPT-4o. The results are averaged over three trials.

As a minor contribution, we create a new benchmark to address some of the drawbacks of existing datasets. Prior datasets feature simple counting tasks, e.g., counting "strawberries", and lack intra-class complexity. To address these issues, we develop a challenging benchmark for counting emoji icons. The subtle variations within emoji classes make this benchmark uniquely difficult.

## 2 Related Work

Early counting models, referred to as class-specific, targeted counting problems for certain categories (Arteta et al., 2016; Babu Sam et al., 2022; Mundhenk et al., 2016; Xie et al., 2018), such as cars, people, or cells. Later, with the emergence of stronger vision models and large-scale datasets, class-agnostic methods were proposed that could count objects from a wide variety of categories. However, most existing class-agnostic, or open-world, models require visual exemplars of the target objects (Đukić et al., 2023; Gong et al., 2022; Lin et al., 2022; Liu et al., 2022; Lu et al., 2019; Nguyen et al., 2022; Ranjan et al., 2021; Shi et al., 2022; Yang et al., 2021; You et al., 2023).

**Text-based counting models.** With the advent of vision-language foundation models such as CLIP and GroundingDINO, text-based open-world models have been proposed that are trained specifically for counting. Leveraging the rich textual and visual feature extraction capabilities of foundation models, obtained through web-scale training, the text-based counting models by Amini-Naieni et al. (2023); Dai et al. (2024); Kang et al. (2024); Amini-Naieni et al. (2024) have started to demonstrate comparable or superior accuracy. In addition, Shi et al. (2024) introduce TFOC, a counting model that does not require any counting-specific training. Instead, they cast the counting problem as a prompt-based segmentation task, using SAM (Kirillov et al., 2023) to obtain segmentation masks that determine the output number. Despite this progress, these models remain constrained by two principal limitations. First, their performance can degrade significantly under considerable distribution shift in the input samples. Second, the prompts they interpret are typically limited to simple object categories or referring expressions; as task complexity increases, these models often fail. Employing LVLMs for counting offers a potential pathway to mitigate these issues.

**Leveraging the concept of divide and conquer for counting.** The concept of divide and conquer has been used in early work (Chattopadhyay et al., 2017; Xiong et al., 2019; Stahl et al., 2018). Chattopadhyay et al. (2017) use an image-level divide and conquer approach and train a convolutional neural network (CNN) that can count objects from a predetermined and limited set of categories in sub-images. Xiong et al. (2019) propose applying the divide step on the convolutional feature map instead of the input image to avoid repeatedly computing convolutional features for each sub-image, thereby improving efficiency. However, the CNN in their work is only capable of counting a single object category. Similar to Chattopadhyay et al. (2017),

Stahl et al. (2018) also employ image-level division and train a CNN to count objects from a predetermined set of categories. Nonetheless, their method does not require local image annotations for training.

**Assessment of LVLMs' counting performance.** Several prior works have explored the visual counting capabilities of LVLMs as part of broader evaluations, underscoring the difficulties these models encounter in counting tasks Yin et al. (2023); Xu et al. (2024); Yang et al. (2023). However, these studies have not focused on developing solutions to address these challenges. In this work, we conduct a comprehensive quantitative assessment of LVLMs' counting performance across diverse visual counting benchmarks. More importantly, we propose a simple yet effective baseline method to improve their ability to count large numbers. Our baseline method is not tailored to any specific model and can be used in a plug-and-play manner with different LVLMs. Regarding the categories outlined earlier, our method, LVLM-Count, is an open-world, prompt-based counting approach that requires no additional training. To the best of our knowledge, we are the first to propose a divide-and-conquer strategy that avoids splitting and potentially double-counting objects of interest specified by an arbitrary text prompt.

## 3  LVLM-Count

Our proposed method aims to answer counting questions by dividing an image into sub-images while avoiding cuts through objects of interest. LVLM-Count consists of four key stages. First, in the "Area Detection" stage, we localize areas containing relevant objects. Second, in the "Target Segmentation" stage, we identify and segment the objects of interest. Third, in the "Object-aware Division" stage, we divide the localized areas into sub-images without cutting through the segmented objects. Finally, the LVLM counts the target objects in each sub-image and aggregates the results. Figure 1 illustrates the workflow of our method, which we detail in the following subsections.

### 3.1  Area Detection

In this part of the pipeline, we assume that we are given a counting question $Q$ along with an image. The question $Q$ contains an expression $E$ that specifies a set of objects of interest. The expression $E$ distinguishes these objects from objects of other categories or the same category but with different attributes present in the image. By employing an LLM, the expression $E$ is extracted from $Q$. For example, let $Q$ be "How many brown eggs are in the image?". $Q$ is given to an LLM, which is prompted to return the expression $E$, "brown eggs", referring to the objects we want to count. After $E$ is extracted, it is provided as input to GroundingDINO along with the image. The output of GroundingDINO may be a single bounding box or a set of bounding boxes that have relevance to $E$ beyond a certain threshold. These bounding boxes often overlap and typically contain repeated objects. Thus, all the overlapping output bounding boxes are merged. After merging, a set of non-overlapping areas of interest may remain. We consider the non-overlapping areas as "detected areas", which are then cropped to be passed to the next stage. This process is illustrated in Figure 3.

### 3.2  Target Segmentation

The cropped images from the first stage contain objects of interest, and the ultimate goal is to divide them without cutting through those objects. However, a prerequisite for implementing such a mechanism is to first detect and localize the objects of interest. Each cropped image is fed into an open-world detection model along with $E$. The output of the open-world detection model produces a bounding box for each object of interest. The bounding boxes are then given as input to a segmentation model, which returns segmentation masks for the objects within each bounding box. We illustrate an example of this process in Figure 4.

**How to determine the bounding boxes.** To determine the bounding boxes, we use GroundingDINO and set the bounding box probability threshold to a low value to avoid missing any object of interest. The bounding boxes alone cannot help with the object-aware division of the detected areas due to their rigid structure, which includes redundant areas in the vicinity of the object and, in the worst case, overlaps with other bounding boxes. Our goal is to precisely locate the pixels of an object of interest.

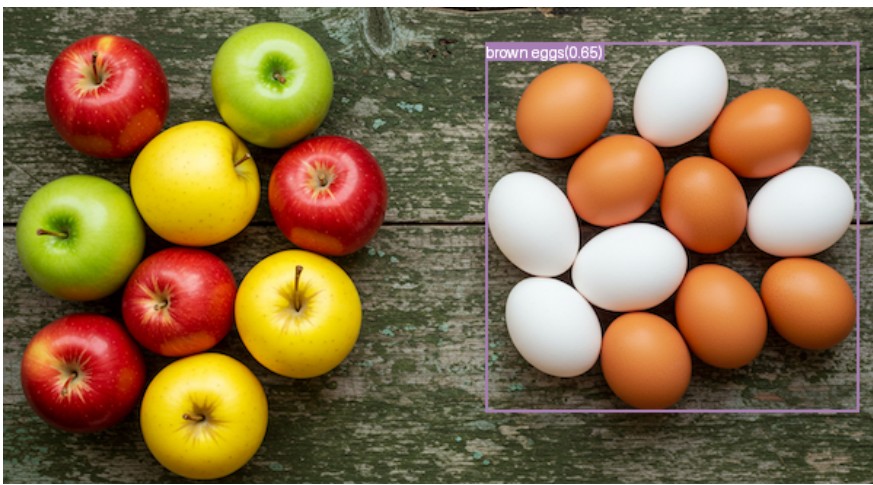

Figure 3: Illustration of the area detection stage of LVLM-Count. For this image, $Q$ is set to "How many brown eggs are in the image". The LLM that is used in this step returns an $E$ which is "brown eggs". $E$ and the original image are given as input to GroundingDINO, which returns a bounding box. If the grounding model returns multiple bounding boxes, they are merged to form the final detected area.

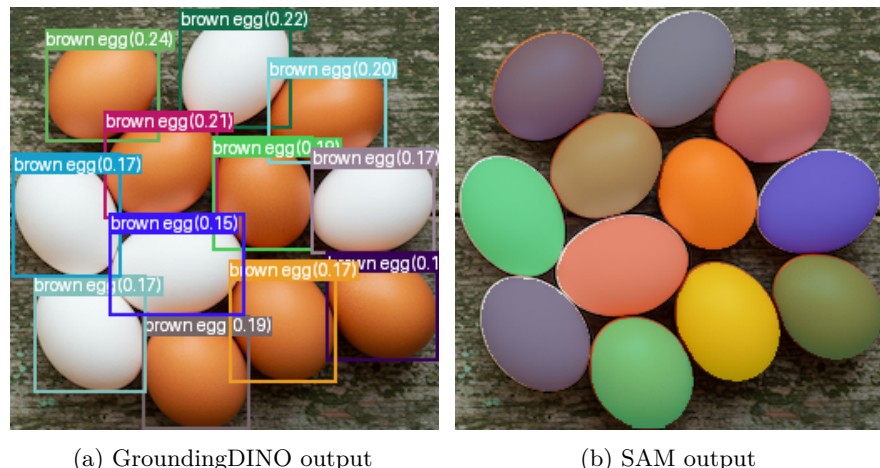

(a) GroundingDINO output          (b) SAM output

Figure 4: Illustration of the target segmentation step of LVLM-Count. The goal is to produce all the instance masks for $E$ set to "brown egg". The cropped detected area from Figure 3, together with $E$, is given as input to GroundingDINO, which produces the output shown in Figure 4a. Figure 4a is then given as input to SAM, which produces the output shown in Figure 4b.

**How to determine the segmentation masks.** We employ a pre-trained segmentation model, specifically SAM (Kirillov et al., 2023), for the segmentation task. This model accepts bounding boxes as prompts and generates masks covering the most prominent objects within these boxes. However, we do not use SAM's output masks directly, as in crowded scenes or cases with multiple occlusions, the masks often overlap, making it difficult to identify reliable division paths. To address this issue, we apply several post-processing steps. First, we perform non-maximum suppression on SAM-generated masks to eliminate those with significant overlaps corresponding to less certain bounding boxes. Additionally, we apply an erosion function to the segmentations, ensuring adjacent masks maintain a minimum separation of two pixels. For a detailed analysis and visual examples of how these post-processing operations enhance robustness in crowded scenes and occlusion cases, see Appendix C. The final processed masks for each cropped area are then passed to the next stage of our pipeline.

**Robustness to the Accuracy of Area Detection and Target Segmentation Stages.** Although we employ GroundingDINO—a state-of-the-art model for grounding and detection—it lacks the flexibility and robustness of LVLMs when generalizing to unseen data and complex concepts. Consequently, we minimizes dependence on the accuracy of the detection model used for area detection and target segmentation. Rather than prioritizing accuracy, we emphasize: i) not missing any region containing objects of interest during area detection, and ii) not missing segmentation masks for any objects of interest during target segmentation. These objectives are easily achieved by setting GroundingDINO's detection threshold to a very low value in both stages.

A low threshold may lead to false positives. In area detection, this could result in regions that contain no objects of interest. However, since counting is performed by the LVLM in the final stage, these regions will be ignored. For target segmentation, false positive masks might occur, but the consequence is that some irrelevant objects will be protected from being cut by the division lines, just as the objects of interest are. Furthermore, as we will demonstrate in subsequent sections, all masks are removed from the sub-images after division, ensuring no noise propagates from this stage to the LVLM-based counting stage.

We demonstrate the effectiveness and robustness of our strategy in the early pipeline stages by showing that LVLM-Count significantly improves LVLMs' performance on one of the most challenging counting datasets—the Penguin Dataset Penguin Research (2016), as detailed in Section 4.2. This dataset is particularly difficult due to frequent occlusion, camouflage, and complex backgrounds Arteta et al. (2016). Furthermore, in our ablation studies, we test a variant of our pipeline that removes the detection model entirely. Instead, SAM is configured to generate segmentation masks for all objects in the scene. As shown in Table 5, our method still enhances counting performance in this configuration, underscoring its robustness and minimal dependence on the detection model during the initial pipeline stages.

### 3.3 Object-aware Division

In this stage, the cropped image is divided into appropriate sub-images so that no object of interest is cut by the dividing paths. The core idea is that the dividing paths should not intersect the pixels covered by the masks corresponding to the objects of interest. This step consists of two sub-steps. First, we decide the starting and ending points of the paths. Second, we draw the paths. Below, we describe how we approach these two sub-steps.

**How to determine the starting and ending points of the paths.** We utilize an unsupervised and non-parametric approach, to obtain the start and end points of the paths. A few pixels are sampled from each of the masks. To determine the location of the division paths on the $x$-axis, the samples taken from the masks are projected onto the $x$-axis. The projected points are automatically clustered using a non-parametric mean-shift algorithm[1] (Comaniciu & Meer, 2002). Once the clusters are identified, the point between the point with the highest $x$ value in one cluster and the point with the lowest $x$ value in the subsequent cluster is considered the $x$-coordinate of a division path. In effect, knowing the $x$-coordinate of a vertical path means that the coordinates of its endpoints are known. In particular, assuming height $h$ for an image crop, we consider $P_s = (x, 0)$ and $P_e = (x, h)$ as the start and end points, respectively. Note that using this technique, we obtain the appropriate coordinates for the division paths, as well as the number of paths. For example, if there is only one cluster along the $x$-axis, no division is required, and if there are two clusters, one vertical path will divide the image into two parts. We illustrate this approach in Figure 5.

**How to draw the paths.** Previous step obtains the endpoints of the division paths. Assume $P_s = (x, 0)$ and $P_e = (x, h)$ are the start and end points of a vertical division path, respectively. In an ideal case where there are no masks in the path of a straight line connecting the two points, this path will be drawn by connecting all the pixels on the straight line. However, there are potential masks that can be considered obstacles blocking the path. In other words, beginning from $P_s$, the line needs to go around these obstacles to reach $P_e$. Consequently, we treat this as a 2-dimensional path-finding problem. To solve the problem, we build a 2D binary map, $I_B$, where the pixels covered by the masks are turned into black, indicating them as obstacles, and all the other pixels are turned into white, showing they are open for passage. This

---

[1]In our implementation, we employ the MeanShift algorithm from the Scikit-learn library (Pedregosa et al., 2011), utilizing its default heuristic for automatic bandwidth estimation.

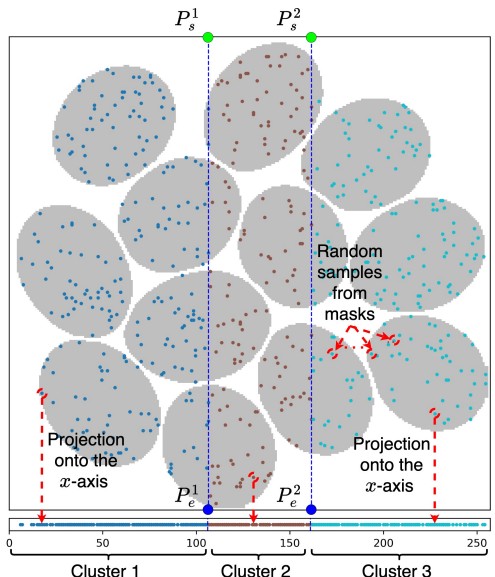

Figure 5: Illustration of the unsupervised and non-parametric method to obtain the division points $(P_s^1, P_e^1)$, and $(P_s^2, P_e^2,)$. A few pixels are sampled (shown as points inside the segmented objects) from the pixels composing target masks. The samples are projected onto the $x$-axis. The projected points are clustered using mean-shift clustering. The point in the middle of two consecutive clusters is considered a vertical division point. Blue lines are solely for illustration

binary image $I_B$ is mapped into a graph $G$, where each white pixel is a node, and it is connected to all of its white neighboring pixels. We use the $A^*(G, P_s, P_e, g)$ search algorithm to find a path that connects $P_s$ to $P_e$, where the heuristic $g$ is set to be Manhattan distance. The output of $A^*$ is a set of connected pixels that go around the obstacles and connect $P_s$ to $P_e$, creating an object-aware division path, as shown in Figure 6. The path-finding algorithm is run for all division coordinates. Finally, we draw the image contours based on these division paths and take the area surrounded by each contour as a resulting sub-image. Note that although not part of the main LVLM-Count pipeline, the above two steps can be applied to the $y$-axis in the same manner to obtain horizontal division paths. For further discussion, the reader may refer to Appendix S.

### 3.4 Target Counting

All the sub-images obtained from the cropped areas are gathered. Then, question $Q$ and each sub-image are given as input to an LVLM. At the end of the loop, the recorded numbers for the sub-images are aggregated to form the final answer. For images with a very large number of objects, sometimes LVLMs refuse to count, citing the large number. In those cases, a new prompt requires the model to give the closest estimate of the number.

## 4 Experiments

In this section, we present the performance results of different LVLMs and their enhanced performance results using our method on a counting-specific dataset, a counting benchmark taken from a popular vision datasets, and a challenging counting benchmark that we propose using emoji icons. Additionally, we show the success of our design in enhancing counting performance on one of the most challenging counting datasets that features heavy occlusions and complex backgrounds. The code to reproduce the experiments is available at our GitHub repository [2]

---

[2]Our GitHub repository is available at: `https://github.com/MuhammadFetrat/lvlm-count`

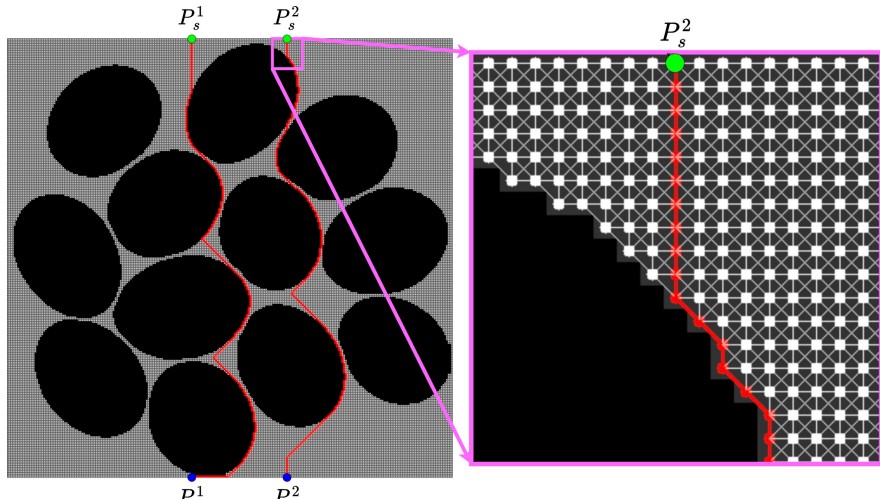

Figure 6: Illustration of object-aware division. The masks are turned into a black-and-white image. A dividing path is found by connecting $P_s$ to $P_e$ using the $A^*$ search algorithm in a graph that corresponds to the binary image. The only nodes in the graph are white pixels of the black-and-white image, which are connected to all other white pixels in their $3 \times 3$ neighborhood. The nodes and edges on the obtained paths have been colored red.

## 4.1 Datasets and Benchmarks

**FSC-147 (Ranjan et al., 2021).** FSC-147 is a counting dataset that contains 6135 images, spanning 147 different object categories such as kitchen utensils, office supplies, vehicles, and animals. The number of objects in each image ranges from 7 to 3731, with an average of 56 objects per image. The dataset is split into training, validation, and test sets. A total of 89 object categories are assigned to the training set, 29 to the validation set, and 29 to the test set, with different categories in each split. The training set contains 3659 images, with the validation and test sets containing 1286 and 1190 images, respectively. For each image in the test set, a single category name is given, and the expected output is the number of instances.

**PASCAL VOC Benchmark** We build a counting benchmark from PASCAL VOC dataset (Everingham et al., 2015). Similar to Chattopadhyay et al. (2017), we choose PASCAL VOC 2007 among other variants. This variant contains a training set of 2501 images, a validation set of 2510 images, and a test set of 4952 images, with 20 object categories that remain consistent across the splits. Each image includes annotations for instances of the 20 object categories in the dataset. We first create 20 simple counting questions asking for the number of objects from each of the 20 categories for every image in the test set. Then, we randomly sample five questions from each ground truth count. This process resulted in 102 questions in total. Finally, we manually checked the ground truth counts and corrected them if required.

**Emoji-Count.** To our knowledge, no counting benchmark exists with large number of objects in a scene involving complex reasoning. To this end, we propose a challenging counting benchmark using emoji icons. From the 1816 standard emoji icons, we remove those that directly overlap with concepts demonstrated by other icons. We then group the remaining 1197 icons into 82 classes. In each class, there are icons from the same or similar object categories, but with subtle differences that require complex reasoning to distinguish. For each of the 82 classes, an empty $1024 \times 1024$ image is first created. This image is filled with up to six categories chosen randomly from the class, with each category having a random count between 30 and 50 in the image. For each image, we create questions that ask the number of instances of the available categories in the image. This results in 415 image-question pairs. We illustrate two examples of this dataset in Figure 7.

**Penguin Benchmark.** The challenging Penguin dataset Penguin Research (2016) consistently exhibits heavy occlusion and complex background patterns that can easily be mistaken for penguins Arteta et al. (2016). The test set of this dataset is quite large, containing thousands of images with penguin counts

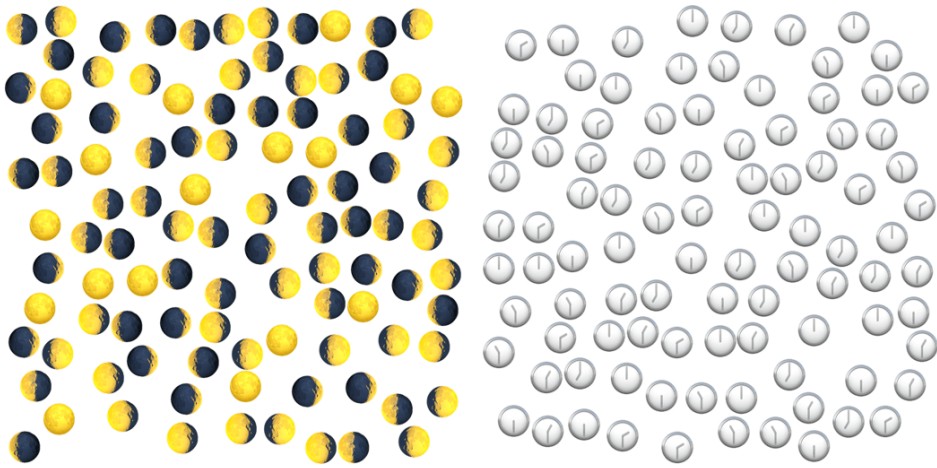

(a) Q: How many waning gibbous moons are there in the image? Answer: 18.

(b) Q: How many clocks at time "two-thirty" are there in the image? Answer: 15.

Figure 7: Illustration of a smaller version of two challenging cases from Emoji-Count. In Figure 7a, the class name is "Moon Phase". In Figure 7b, the class name is "Clock Time".

ranging from 0 to 213. To build a manageable benchmark, we randomly sample 100 images while preserving a balanced ground truth range. We refer the reader to Appendix D for more details about this benchmark.

### 4.2 Results

This section presents the numerical results of our experiments with the base LVLMs and their corresponding LVLM-Count on each benchmark described in Section 4.1. In the experiments, we consider two baselines: (i) a baseline where the number of target segmentation masks is taken as the final answer, and (ii) a baseline where an image of the generated segmentation masks is provided to GPT-4o along with a prompt to count them. For reference, we also report the performance of three state-of-the-art (SOTA) text-based counting-specific models: GroundingREC (Dai et al., 2024), CountGD (Amini-Naieni et al., 2024), and DAVE$_{prm}$ (Pelhan et al., 2024).

It is important to note that the primary aim of this work is to propose a method that enhances the counting ability of an arbitrary large vision-language model (LVLM) across diverse datasets. The inclusion of these specialized models in our comparison should not be interpreted as an attempt to replace or compete with them on specific counting tasks—particularly those involving simple counting. These models are trained on dedicated counting datasets and perform efficiently on test samples from the same distribution or datasets with high category overlap. However, their performance may degrade significantly when applied to out-of-distribution samples. Furthermore, these models are constrained by the type and complexity of the input prompts they can process; they typically only accept simple category names or referring expressions, and often fail when presented with complex phrases or sentences.

In contrast, our method is designed to be comprehensive. Although it may not match the efficiency of task-specific models on datasets they were trained on, it performs robustly across a wide range of test distributions, despite not being explicitly trained on any of the tasks. Moreover, it is capable of handling arbitrary prompt types and structures, regardless of complexity. We also report the performance of TFOC (Shi et al., 2024), a SOTA training-free model. In essence, TFOC is a segmentation-based counting approach, similar to our baselines, but employs a more sophisticated methodology.

The above discussion is reflected in our experimental results. For the trained counting models, we report performance using weights obtained from training on the FSC-147 training set. These models perform well on the FSC-147 dataset and on the PASCAL VOC benchmark, which has high category overlap with FSC-147. However, on the Emoji-Count dataset—which exhibits a different distribution and requires complex

understanding—their performance drops considerably. In contrast, our method, LVLM-Count, maintains high performance even on challenging benchmarks such as Emoji-Count. Another experiment on the TallyQA-Complex benchmark, which contains real-world scenes with complex counting questions, further reveals the limited complex reasoning capability of conventional trained models. LVLMs and our proposed LVLM-Count, however, demonstrate a superior ability to handle such challenging cases. The corresponding results on the TallyQA-Complex benchmark are provided in Appendix T.

We also conduct an ablation study on the effectiveness of different components of our pipeline in Appendix A. In summary, the ablation experiments support three conclusions: (i) each component has a positive effect over the baseline LVLM, and when combined, the full pipeline in Figure 1 achieves the best performance compared to different variants; (ii) naive division with straight lines cannot replace object-aware division; and (iii) LVLM-Count is robust to the accuracy of the target segmentation stage. For visual examples of LVLM-Count's performance on the benchmarks introduced in Section 4.1, see Appendix B. An inference time analysis is provided in Appendix J. This analysis shows that the largest portion of the time is spent querying the LVLM, which is a step common to both the base LVLM and LVLM-Count. The other steps contribute a much smaller share to the overall inference time.

**FSC-147.** We evaluate the performance of various LVLMs and their enhanced performance using our method, LVLM-Count, on the test set of the FSC-147 dataset. We consider two baselines: (i) the baseline where the number of target segmentation masks is taken as the final answer, and (ii) the baseline where an image of the generated segmentation masks is provided to GPT-4o with a prompt to count the number of masks. Our experiments involve GPT-4o (a leading proprietary model), as well as two open-source models: Qwen2 VL 72B AWQ (Yang et al., 2024) and Gemma 3 27B Team et al. (2025). The expression $E$ used in different stages of our method corresponds to the category name provided in the test set. A simple query $Q$ in the form of "How many $E$ are there?" is constructed and used as the text prompt for the LVLM during the counting stage. In all experiments, the detection thresholds are set to 0.1. The results are presented in Table 1, reporting the mean absolute error (MAE) and root mean square error (RMSE). Our findings indicate that LVLM-Count improves the performance of all three LVLMs in terms of MAE. Interestingly, while the base Qwen2 and Gemma 3 models are much less powerful than their commercial counterpart, they outperform the base GPT-4o when integrated into our pipeline.

Table 1: Evaluation on the test set of the FSC-147 dataset. In all the tables of this section, the results for base LVLMs and LVLM-Count are reported over six trials. In the tables, columns marked with $\Delta$ show the performance difference between the LVLM-Count and the base LVLM it uses. Green indicates improvement, while red represents degradation. Also, the 95% confidence intervals for the MAEs are reported in the column named MAE 95% CI. To see the measured accuracy metrics for this dataset and the subsequent benchmarks, refer to Appendix I. Additionally, MAE analysis across different intervals of ground truth values for FSC-147 is provided in Appendix H.

| Method | Trained Model | MAE ↓ | $\Delta$ | MAE 95% CI | RMSE ↓ | $\Delta$ |
|---|---|---|---|---|---|---|
| TFOC (Shi et al., 2024) | ✗ | 24.79 | - | - | 137.15 | - |
| DAVE$_{prm}$ (Pelhan et al., 2024) | ✓ | 14.90 | - | - | 103.42 | - |
| CountGD (Amini-Naieni et al., 2024) | ✓ | 14.76 | - | - | 120.42 | - |
| GroundingREC (Dai et al., 2024) | ✓ | 10.12 | - | - | 107.19 | - |
| Number of target segmentaion masks | ✗ | 44.14 | - | - | 154.39 | - |
| GPT-4o counting target segmentaion masks | ✗ | 38.45 | - | - | 156.11 | - |
| GPT-4o | ✗ | 25.57 | - | [24.74, 26.39] | 137.26 | - |
| LVLM-Count (GPT-4o as LVLM) | ✗ | 17.86 | ↓ 7.71 | [16.96, 18.77] | 91.71 | ↓ 45.55 |
| Gemma 3 27B | ✗ | 30.59 | - | [30.22, 30.97] | 132.61 | - |
| LVLM-Count (Gemma 3 27B as LVLM) | ✗ | 20.25 | ↓ 10.34 | [19.62, 20.89] | 110.45 | ↓ 22.16 |
| Qwen2 VL 72B AWQ | ✗ | 34.18 | - | [32.85, 35.51] | 149.49 | - |
| LVLM-Count (Qwen2 VL 72B AWQ as LVLM) | ✗ | 22.29 | ↓ 11.89 | [21.63, 22.96] | 119.46 | ↓ 30.44 |

**PASCAL VOC Benchmark.** We evaluate the performance of three LVLMs—GPT-4o, Qwen2, and Gemma 3—on this benchmark, both with and without LVLM-Count. As shown in Table 2, LVLM-Count consistently outperforms the base LVLMs.

Table 2: Evaluation on the PASCAL VOC counting benchmark.

| Method | MAE ↓ | Δ | MAE 95% CI | RMSE ↓ | Δ |
|---|---|---|---|---|---|
| TFOC (Shi et al., 2024) | 12.03 | - | - | 18.18 | - |
| DAVE$_{prm}$ (Pelhan et al., 2024) | 12.39 | - | - | 22.81 | - |
| CountGD (Amini-Naieni et al., 2024) | 2.81 | - | - | 7.01 | - |
| GroundingRec (Dai et al., 2024) | 4.05 | - | - | 7.80 | - |
| Number of the target segmentation masks | 4.03 | - | - | 7.47 | - |
| GPT-4o counting target segmentaion masks | 6.60 | - | - | 15.09 | - |
| GPT4o | 4.64 | - | [4.45, 4.82] | 8.56 | - |
| LVLM-Count (GPT4o as LVLM) | 3.42 | ↓ 1.22 | [3.25, 3.59] | 7.17 | ↓ 1.39 |
| Gemma 3 27B | 3.40 | - | [3.24 ,3.55] | 7.54 | - |
| LVLM-Count (Gemma 3 27B as LVLM) | 2.97 | ↓ 0.43 | [2.82, 3.11] | 6.16 | ↓ 1.38 |
| Qwen2 VL 72B AWQ | 4.80 | - | [4.61, 4.99] | 8.71 | - |
| LVLM-Count (Qwen2 VL 72B AWQ as LVLM) | 4.12 | ↓ 0.68 | [3.87, 4.37] | 7.76 | ↓ 0.95 |

**Emoji-Count.** We evaluate the performance of the LVLMs and their performance using LVLM-Count on the Emoji-Count benchmark. The results are shown in Table 3. This is a challenging benchmark, as it requires understanding complex concepts. We observe that taking the number of masks as the final count performs particularly poorly, as for any object of interest in the image, the segmentation stage tends to segment all the objects and cannot distinguish between different icons. Although GPT-4o and Gemma 3 show reasonable performance, the other open-source model does not perform well. Nonetheless, the performance of all three base LVLMs is significantly enhanced by LVLM-Count, especially Qwen2, which performs almost on par with GPT-4o when LVLM-Count is used for it.

Table 3: Evaluation on the Emoji-Count benchmark.

| Method | MAE ↓ | Δ | MAE 95% CI | RMSE ↓ | Δ |
|---|---|---|---|---|---|
| TFOC (Shi et al., 2024) | 64.64 | - | - | 87.45 | - |
| DAVE$_{prm}$ (Pelhan et al., 2024) | 198.99 | - | - | 208.08 | - |
| CountGD (Amini-Naieni et al., 2024) | 137.93 | - | - | 156.80 | - |
| GroundingREC (Dai et al., 2024) | 143.22 | - | - | 158.74 | - |
| Number of the target segmentation masks | 82.47 | - | - | 107.98 | - |
| GPT-4o counting target segmentaion masks | 107.72 | - | - | 162.12 | - |
| GPT-4o | 23.57 | - | [22.37, 24.76] | 36.97 | - |
| LVLM-Count (GPT-4o as LVLM) | 16.57 | ↓ 7 | [16.25, 16.90] | 33.11 | ↓ 3.86 |
| Gemma 3 27B | 21.39 | - | [21.32, 21.45] | 24.04 | - |
| LVLM-Count (Gemma3 27B as LVLM) | 16.16 | ↓ 5.23 | [15.76, 16.56] | 21.27 | ↓ 2.77 |
| Qwen2 VL 72B AWQ | 78.05 | - | [72.17, 83.93] | 159.22 | - |
| LVLM-Count (Qwen2 VL 72B AWQ as LVLM) | 24.43 | ↓ 53.62 | [23.65, 25.21] | 43.38 | ↓ 115.84 |

**Penguin Benchmark.** We report the counting performance on the Penguin benchmark in Table 4. We evaluate two variants of LVLM-Count. The primary variant uses GroundingDINO in both the area detection and target segmentation stages. The alternative variant does not use a detection model; instead, SAM is configured to segment all entities in the image. Both variants improve MAE across all three LVLMs, highlighting LVLM-Count's robustness in scenarios with heavy occlusion and complex backgrounds—conditions that pose significant challenges for area detection and target segmentation. Additional details and a visual example showing the segmentation masks and division paths generated in both variants for a sample from the Penguin benchmark can be found in Appendix D.

Table 4: Evaluation on the Penguin benchmark.

| Method | MAE ↓ | Δ | MAE 95% CI | RMSE ↓ | Δ |
|---|---|---|---|---|---|
| GPT4o | 35.18 | - | [33.13, 37.23] | 45.76 | - |
| LVLM-Count( Main variant, using GPT-4o ) | 26.76 | ↓ 8.42 | [25.86, 27.66] | 38.60 | ↓ 7.16 |
| LVLM-Count(SAM-only variant, using GPT-4o) | 29.02 | ↓ 6.16 | [28.21, 29.84] | 44.89 | ↓ 0.87 |
| Gemma 3 27B | 49.44 | - | [48.94, 49.95] | 60.20 | - |
| LVLM-Count( Main variant, using Gemma 3 ) | 34.13 | ↓ 15.31 | [31.72, 36.53] | 44.11 | ↓ 16.09 |
| LVLM-Count( SAM-only variant, using Gemma 3 ) | 41.09 | ↓ 8.35 | [39.92, 42.25] | 53.01 | ↓ 7.19 |
| Qwen2 VL 72B AWQ | 44.02 | - | [40.45, 47.60] | 65.59 | - |
| LVLM-Count( Main variant, using Qwen2 ) | 28.21 | ↓ 15.81 | [26.48, 29.94] | 40.40 | ↓ 25.19 |
| LVLM-Count(SAM-only, using Qwen2 ) | 33.33 | ↓ 10.69 | [30.45, 36.21] | 47.55 | ↓ 18.04 |

## 5 Limitations

In this work, we quantitatively evaluated the visual counting performance of several LVLMs on multiple datasets. More importantly, we introduced a simple yet effective baseline method, LVLM-Count, that enhances the visual counting capabilities of LVLMs across the evaluated benchmarks. However, like any method, LVLM-Count has limitations. One limitation arises in cases where sub-images contain no objects of interest: the LVLM may occasionally predict a non-zero value. This reflects a broader weakness in LVLMs, and addressing it will require targeted improvements to ensure their accurate zero prediction in such scenarios. Another limitation occurs with open-source models during the target counting stage. Ideally, the LVLM's output for each sub-image should be a numerical value. While top proprietary LVLMs, such as GPT-4o, offer functionalities like JSON schema to enforce structured responses (e.g., ensuring a numerical output), open-source models typically lack such features. For these models, the prompt must include explicit instructions to format the response correctly. For instance, an instruction like "Place the final predicted number inside [[]]" enables the use of regex searches to extract the number for aggregation.

## Acknowledgments

K. Fountoulakis would like to acknowledge the support of the Natural Sciences and Engineering Research Council of Canada (NSERC). Cette recherche a été financée par le Conseil de recherches en sciences naturelles et en génie du Canada (CRSNG), [RGPIN-2019-04067, DGECR-2019-00147].

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

# Appendix Table of Contents

## A    Ablation Study

We examine the effect of each stage in our method: (i) area detection and (ii) object-aware division using masks produced during target segmentation. We design different experiments to investigate each stage's individual contribution, as well as their combined effect in our pipeline.

To demonstrate our method's minimal dependence on detection accuracy in the area detection and target segmentation stages, we conduct an experiment without area detection and without using GroundingDINO for segmentation masks. Instead, we configure SAM in "segment anything" mode, which generates masks for all entities in the scene. The results show that our method remains effective even without any detection model, confirming that LVLM-Count has minimal dependence on the accuracy of these initial stages.

Additionally, we conduct an experiment where both area detection and segmentation stages are excluded. In this case, object-aware division cannot be performed, so we divide images using equidistant straight lines into subimages (which we call "naive division"). We also run another experiment with area detection but without segmentation, applying naive division to the detected areas. The results demonstrate that unlike object-aware division, the naive approach is unreliable due to repetitive counting of fragmented objects.

We perform ablation studies using GPT-4o, conducting experiments on the FSC-147 test set. Table 5 presents the results of these ablation experiments.

Table 5: An ablation study of LVLM-Count on the FSC-147 test dataset. Columns marked with $\Delta$ show the performance difference between LVLM-Count and the base LVLM. Green indicates improvement, while red represents degradation.

| Method | MAE ↓ | $\Delta$ | RMSE ↓ | $\Delta$ |
|---|---|---|---|---|
| GPT-4o | 25.57 | - | 137.26 | - |
| GPT-4o + Naive division | 33.04 | ↑ 7.47 | 116.83 | ↓ 20.43 |
| GPT-4o + Area Detection + Naive division | 32.69 | ↑ 7.12 | 102.41 | ↓ 34.85 |
| GPT-4o + Area detection | 23.08 | ↓ 2.49 | 120.06 | ↓ 17.2 |
| GPT-4o + Object-aware division (using SAM without GroundingDINO) | 21.01 | ↓ 4.56 | 135.04 | ↓ 2.22 |
| GPT-4o + Object-aware division | 19.17 | ↓ 6.40 | 120.61 | ↓ 16.65 |
| GPT-4o + Area detection + Object-aware division, (equiv. to LVLM-Count) | **17.86** | ↓ 7.71 | **91.71** | ↓ 45.55 |

## B    Visual Examples of LVLM-Count's Performance on the FSC-147 Dataset, PASCAL VOC, and Emoji-Count Benchmarks

This section presents several visual examples showcasing the performance of LVLM-Count on the FSC-147 dataset , PASCAL VOC, and the Emoji-Count benchmarks. The LVLM used in the pipeline to generate these visual examples is GPT-4o.

Figure 8 illustrates three examples of LVLM-Count's performance on FSC-147. Additionally, Figure 9 includes an example from the PASCAL VOC benchmark. Finally, we present three visual examples from the Emoji-Count benchmark in Figure 10. In these visual examples, LVLM-Count achieves more accurate results compared to the base GPT-4o.

## C    Robustness to Crowded Scenes and Heavy Occlusion

One of the key strengths of our approach is its robustness in crowded scenes and images with heavy occlusions. To achieve this robustness, the masks produced by SAM are not directly used for object-aware division. Instead, several important post-processing operations are applied beforehand. The most critical of these is non-maximum suppression, which removes lower-confidence masks that overlap with others. Additionally, an

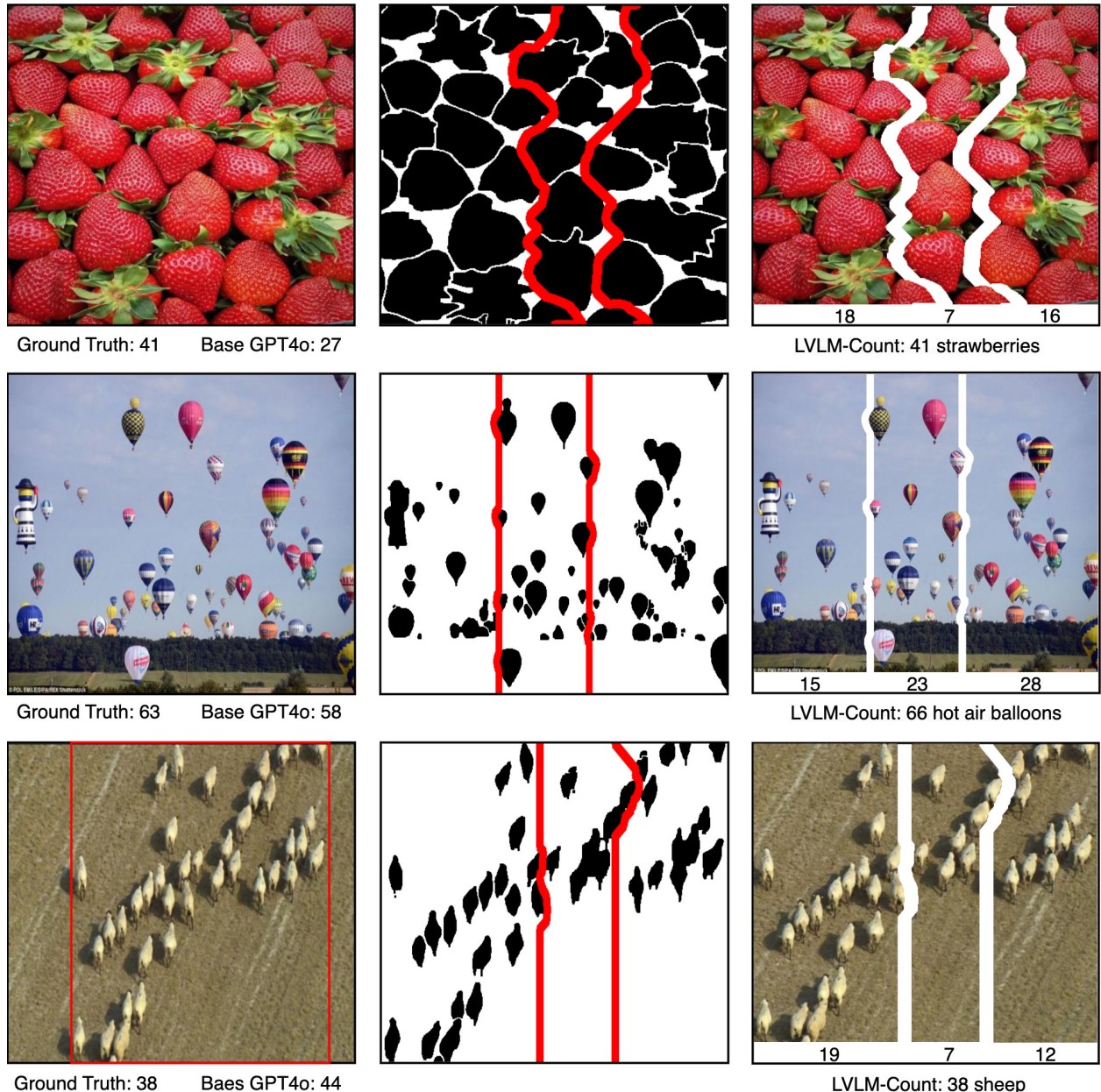

Figure 8: Illustration of three examples of the performance of LVLM-Count on FSC-147. Top row: The object of interest is "strawberry". Middle row: The object of interest is "hot air balloon". Bottom row: The object of interest is "sheep".

erosion function is applied to trim the outermost layer of each mask, ensuring at least two pixels of empty space between adjacent masks to allow for reliable placement of division lines. A polishing step further refines the masks by smoothing their surfaces and eliminating artifacts, preventing division lines from mistakenly passing through them.

Figure 11 demonstrates the strong performance of our method on a randomly selected image from the web containing various types of flowers. This scene features extreme occlusion; nonetheless, our method effectively handles the division task. It is important to note that the strong performance of our method in this challenging scenario is not coincidental. Figure 12 illustrates the impact of our post-processing operations. The upper

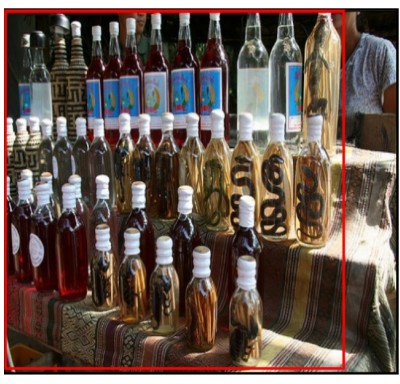 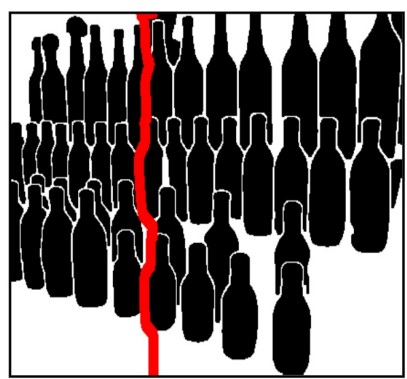 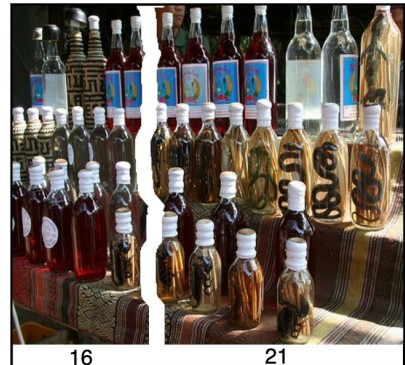

Ground Truth: 41     Base GPT4o: 27                              LVLM-Count: 37 bottles

Figure 9: An example of the performance of LVLM-Count on the PASCAL VOC benchmark. The object of interest is "bottle".

row displays the unprocessed SAM masks, while the bottom row shows the results after post-processing, which generates reliable paths for the division lines.

To further demonstrate the effectiveness of our method in crowded scenes, we present Table 6, which compares the performance of different LVLMs on FSC-147 dataset samples where the ground truth count is 100 or higher. The table contrasts their baseline performance with their performance when using our LVLM-Count method. The results show significant improvements when LVLM-Count is employed. Furthermore, Figure 13 presents three visual examples of LVLM-Count's performance on crowded images from FSC-147.

Table 6: Evaluation on samples with ground truth counts equal to or higher than 100 from the test set of the FSC-147 dataset. Showing the effectiveness of our method for crowded scenes.

| Method | MAE ↓ | Δ | RMSE ↓ | Δ |
|---|---|---|---|---|
| GPT-4o | 89.22 | - | 291.34 | - |
| LVLM-Count (GPT-4o as LVLM) | 57.87 | ↓ 31.35 | 185.29 | ↓ 106.05 |
| Gemma 3 27B | 125.69 | - | 330.38 | - |
| LVLM-Count (Gemma 3 27B as LVLM) | 71.68 | ↓ 54.01 | 251.48 | ↓ 78.9 |
| Qwen2 VL 72B AWQ | 102.64 | - | 302.23 | - |
| LVLM-Count (Qwen2 VL 72B AWQ as LVLM) | 79.66 | ↓ 22.98 | 289.47 | ↓ 12.76 |

## D   Additional Information and Visual Examples from the Penguin Benchmark

To demonstrate the effectiveness of our strategy of setting a low detection threshold to overcome challenges in area detection and target segmentation stages, we evaluate it on a benchmark taken from the Penguin dataset (Penguin Research, 2016). The goal in this dataset is to count penguins in images. This is challenging since images in the dataset consistently exhibit heavy occlusion and complex background patterns that can easily be mistaken for penguins Arteta et al. (2016). This dataset consists of two splits: i) the mixed-site split, in which images from the same camera can appear in both the training and testing sets, and ii) the separate-site split, in which images in each set strictly belong to different cameras. Images in this dataset are annotated by multiple annotators, where each annotator might identify a different number of penguins due to the challenges in locating them within the images. Since annotators usually undercount the penguins, similar to Arteta et al. (2016), we take the maximum number of penguins among the annotations as the ground truth and calculate MAE and RMSE with respect to this value. For additional details regarding the dataset and metric calculations, we direct readers to Arteta et al. (2016).

We construct our benchmark using the separate-site split. Given the dataset's substantial size, which comprises tens of thousands of images, we randomly select 100 samples from the chosen split. To ensure

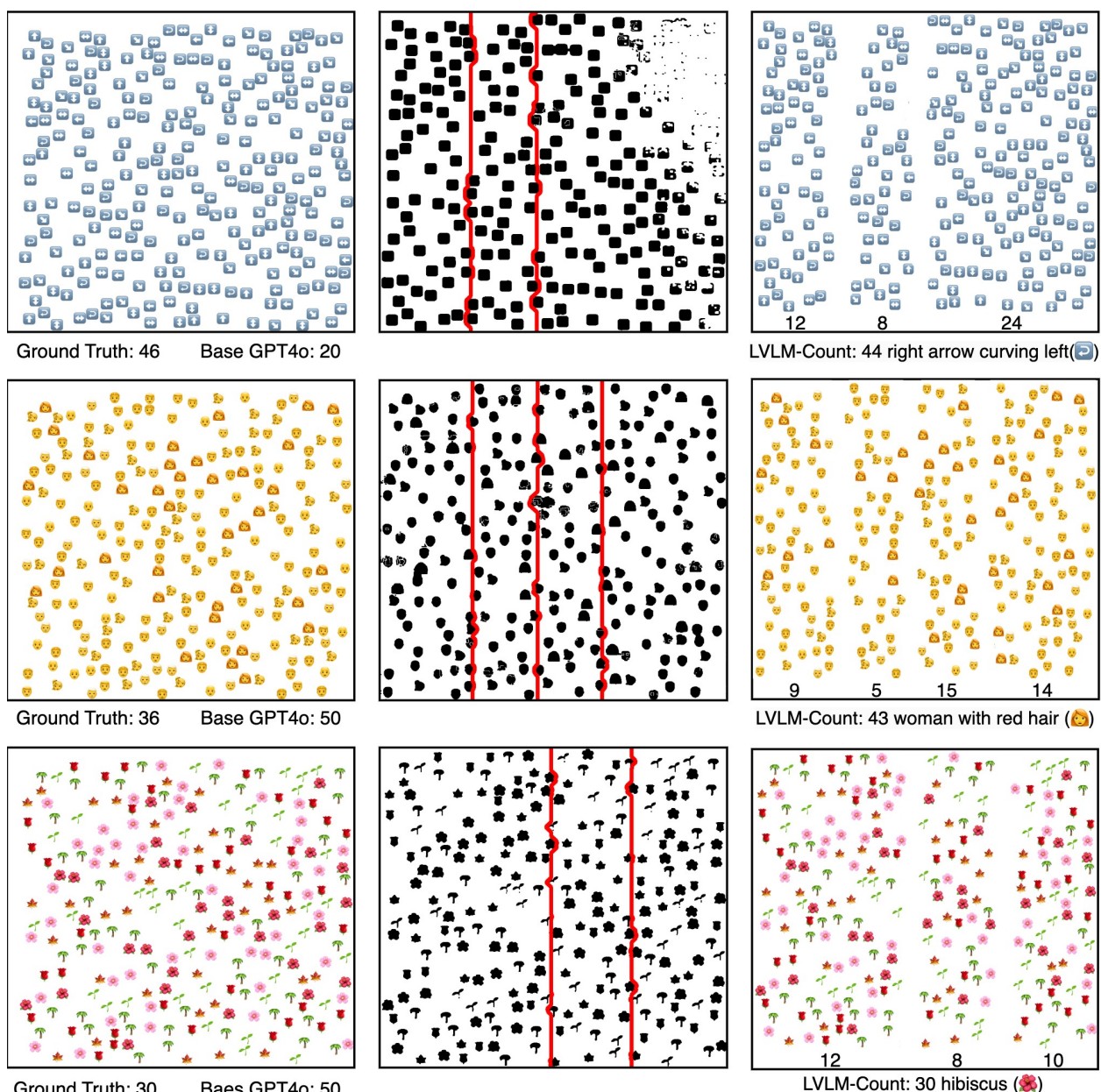

Figure 10: Three examples of the performance of LVLM-Count on the Emoji-Count benchmark. Top row: The object of interest is "right arrow curving left". Middle row: The object of interest is "woman with red hair". Bottom row: The object of interest is "hibiscus".

balance, the sampling probability for an image with a specific ground-truth annotation count is inversely proportional to the frequency of that count in the entire split. We evaluate two variants of our pipeline. The main variant employs GroundingDINO with a low threshold for area detection and target segmentation, while the alternative variant does not use GroundingDINO at all. In this variant, masks are obtained by configuring SAM to segment any entity in the image. The results presented in Table 4 show that both variants significantly improve the counting performance of the LVLMs, demonstrating the robustness of the initial stages in our pipeline.

Figure 14 demonstrates the segmentation masks and division paths for the two variants on a sample from the Penguin benchmark. In Figure 14 (a), GroundingDINO with a low detection threshold is employed.

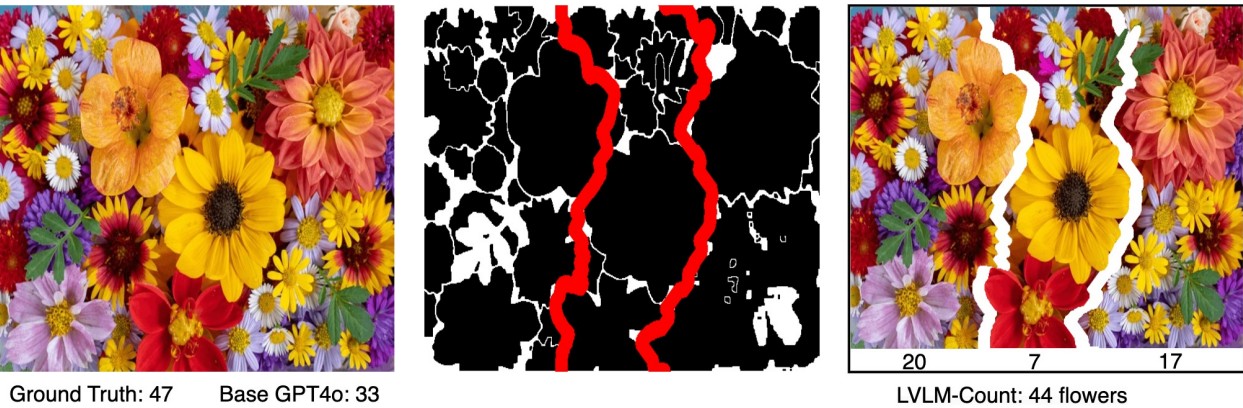

Figure 11: An example of the performance of LVLM-Count on a random crowded image from the web, involving heavy occlusion. The object of interest is "flower".

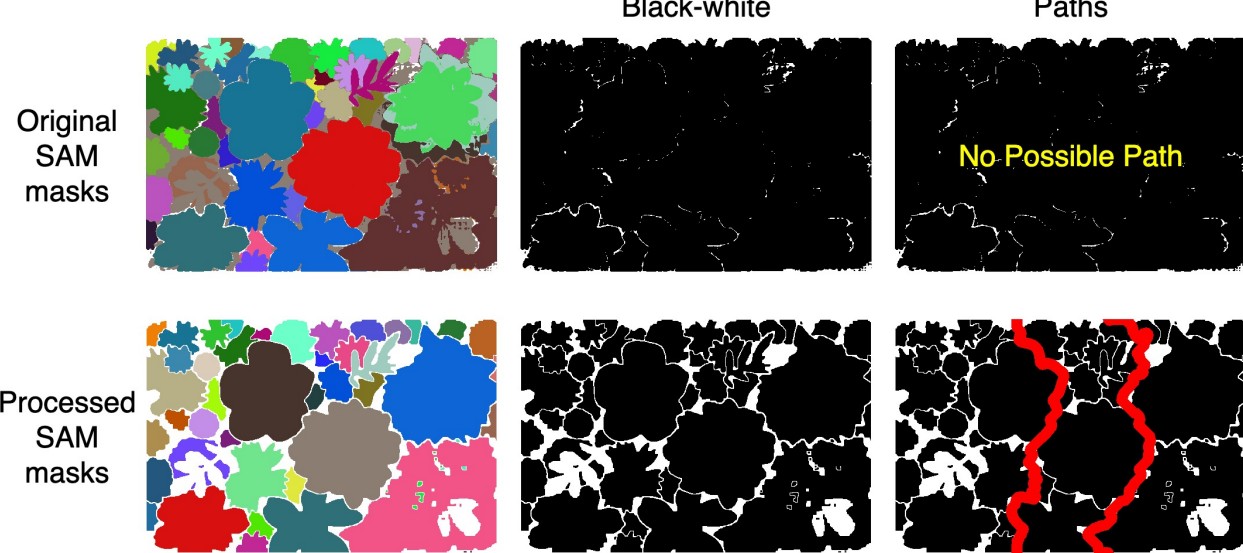

Figure 12: The effect of post-processing on the masks produced by SAM in making LVLM-Count robust to crowded scenes with heavy occlusions. The object of interest is "flower".

It correctly identifies all penguin instances, though there are some false positives. These detections are subsequently segmented by SAM. Figure 14 (c) depicts the division paths found using these masks. In Figure 14 (b), GroundingDINO is not used, and SAM is configured to segment all entities in the scene. As a result, in addition to the penguins, a large number of other objects are segmented. Nevertheless, Figure 14 (d) shows that the division paths derived from these masks successfully partition the image without cutting any penguins. Note that some excessively large masks generated by SAM have been removed due to post-processing in our pipeline.

## E   Real-world Application of LVLM-Count

As stated in Section 1, counting has numerous real-world applications, including but not limited to biology, health, industry, warehousing, and environmental monitoring. Below, we demonstrate the performance of LVLM-Count on examples from the following areas: i) biology/health, ii) industry/warehousing, and iii) environmental monitoring. We also compare its results with those of the base LVLM (GPT-4o for the figures

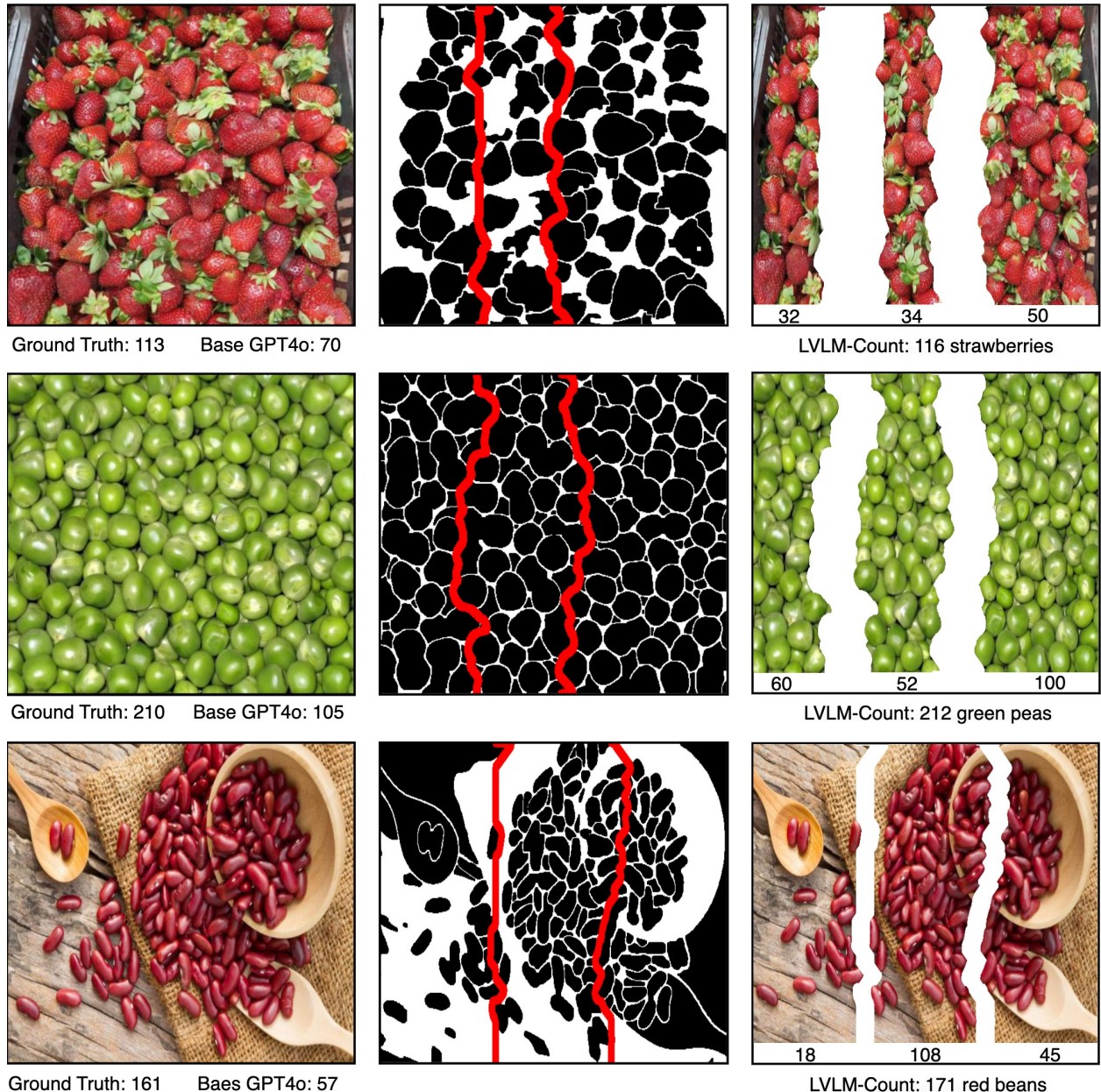

Figure 13: Visual examples of the performance of LVLM-Count on three crowded scenes from the FSC-147 dataset.

in this section). Note that in all examples, the cluster-based approach automatically determines the start and end points of the division paths.

In Figure 15, images of two laboratory samples are analyzed using LVLM-Count. The first row shows an image from a dataset introduced by Lempitsky & Zisserman (2010), which contains simulated bacterial cells from fluorescence-light microscopy, created by Lehmussola et al. (2007). The second row shows an image from the BM dataset introduced by Kainz et al. (2015), which contains bone marrow samples from eight healthy individuals. The standard staining procedure highlights the nuclei of various cell types in blue, while other cellular components appear in shades of pink and red (Paul Cohen et al., 2017). As observed, LVLM-Count achieves much higher accuracy in counting bacterial cells and bone marrow nuclei in the top and bottom rows of Figure 15, respectively, compared to the base LVLM, particularly for the bone marrow nuclei.

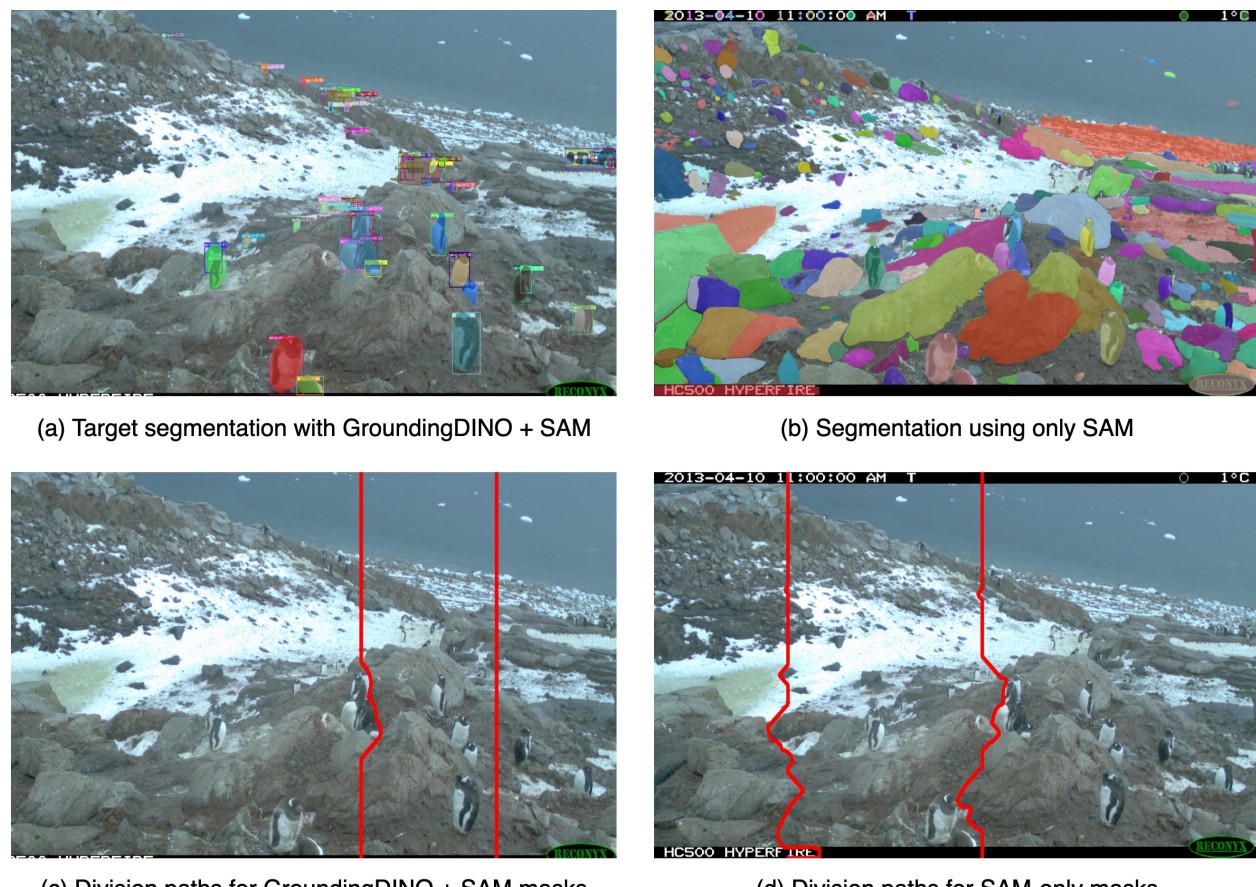

(a) Target segmentation with GroundingDINO + SAM

(b) Segmentation using only SAM

(c) Division paths for GroundingDINO + SAM masks

(d) Division paths for SAM-only masks

Figure 14: Illustration of robustness in detection during the initial pipeline stages: an example from the challenging Penguin benchmark. (a) GroundingDINO with a low threshold successfully detects all penguins, though a few false positives remain. These detections are then segmented by SAM. (b) Masks produced without a detection model: Here, SAM is set to segment all entities in the image. Excessively large masks are filtered out during post-processing in our pipeline. (c) Division paths are derived using the masks generated by GroundingDINO and SAM. (d) Division paths based on SAM-only segmentation effectively partition the image without cutting through any penguins. Note that the slight difference in the area of the images on the left is due to the existence of area detection stage.

In Figure 16, two images from industrial scenes are analyzed using LVLM-Count. The top row shows a sectional image of a stockpile of tree logs, and the bottom row shows an image from an industrial area containing barrels of various colors. For the top image, the objects of interest are the tree logs, while for the bottom image, LVLM-Count is tasked with counting the *blue* barrels. In both cases, LVLM-Count's predictions are significantly closer to the ground truth values than those of the base LVLM, particularly for the tree logs, where the ground truth number is too large for the base LVLM to estimate accurately.

Figure 17 shows an image sourced from a dataset (Penguin Research, 2016) created as part of an ongoing initiative to monitor the penguin population in Antarctica. This dataset comprises images captured hourly by a network of fixed cameras installed at over 40 locations. Over several years, this effort has accumulated over 500,000 images. Zoologists use these images to identify trends in penguin population sizes at each site, facilitating studies on potential correlations with factors such as climate change. Thus, determining the number of penguins in each image is crucial. Given the challenges of engaging human annotators to process such a vast dataset, automating the counting task is highly desirable (Arteta et al., 2016). LVLM-Count is prompted to count the number of penguins in the image, and as observed, its predictions are significantly closer to the ground truth than those of the base LVLM.

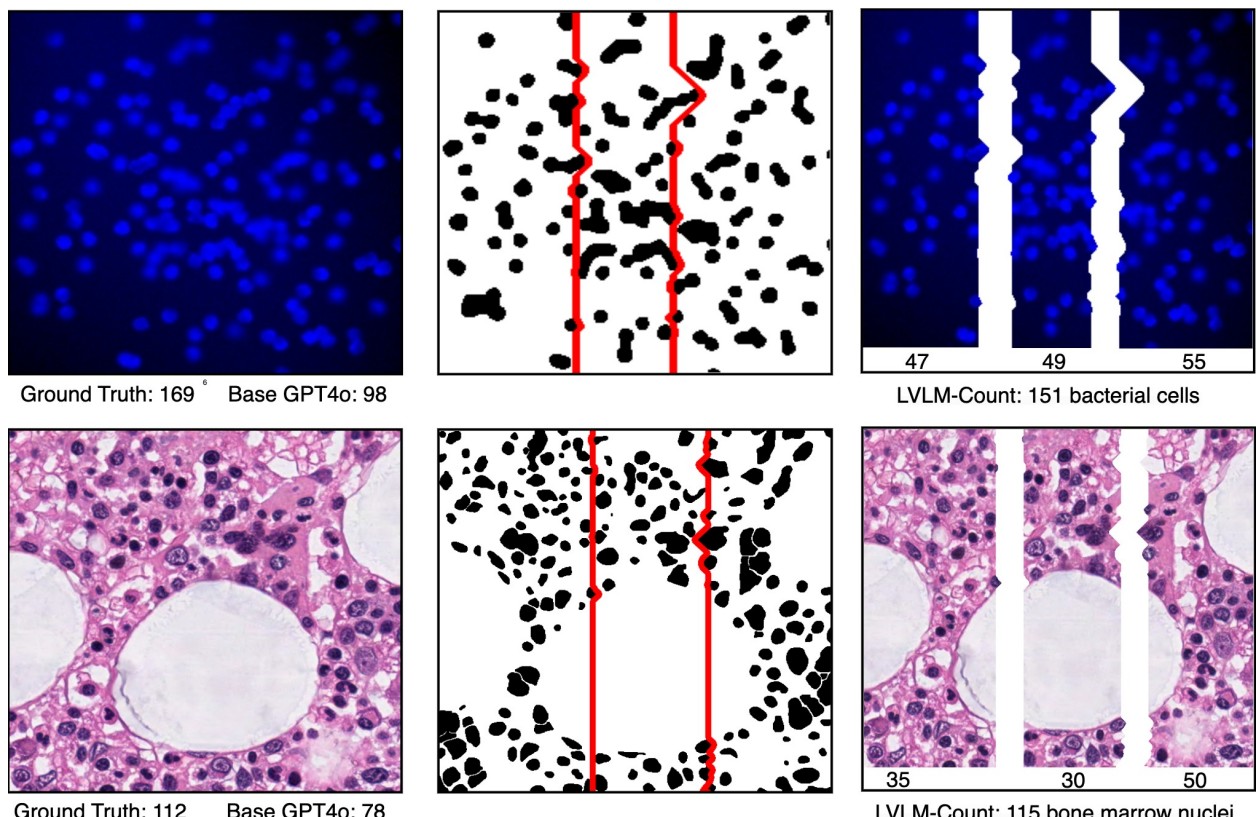

Figure 15: Performance of LVLM-Count on real-world applications in biology/health. The top row shows an image of simulated bacterial cells from fluorescence-light microscopy (Lempitsky & Zisserman, 2010), with the objects of interest being "bacterial cells." The bottom row shows an image of bone marrow, with the nuclei of various cell types highlighted in blue (Kainz et al., 2015), and the objects of interest being "bone marrow nuclei."

## F    LVLM-Count's Power in Handling Multiple Object Categories in the Same Image

LVLM-Count is a highly effective method for handling counting tasks that involve multiple objects in the same image. Its strength in such scenarios stems from the capabilities of LVLMs to answer numerous visual questions about an image and its objects. Depending on the given text prompt, it can count instances of a single object category among others or instances of multiple object categories simultaneously. In this section, we demonstrate how LVLM-Count performs in counting different objects of interest, determined simply by a prompt, using an image with multiple object categories.

The image in Figure 18 contains three object categories: person, cow, and horse. In the top row, the object of interest is "cow." We prompt LVLM-Count to count the cows. First, the masks are produced through the initial stages of our pipeline, and then the cluster-based approach is used to automatically determine the start and end points of the division paths. It can be observed that horses have also been masked as cows. Nonetheless, this does not negatively impact the final answer; it merely causes the division lines to avoid cutting through the horses as well. The counting in LVLM-Count is performed by an LVLM (GPT-4o in this figure) and does not rely on the masks. We observe that GPT-4o successfully counts the number of cows in the resulting subimages, leading to the correct final answer.

In the middle row of Figure 18, the object of interest is "person." LVLM-Count again successfully counts the number of people accurately. A more interesting case is the bottom row of Figure 18, where both cows and persons are objects of interest. We prompt LVLM-Count to count the number of "cows and persons." Similar to the first row of the figure, there are false positive masks here as well. However, LVLM-Count successfully

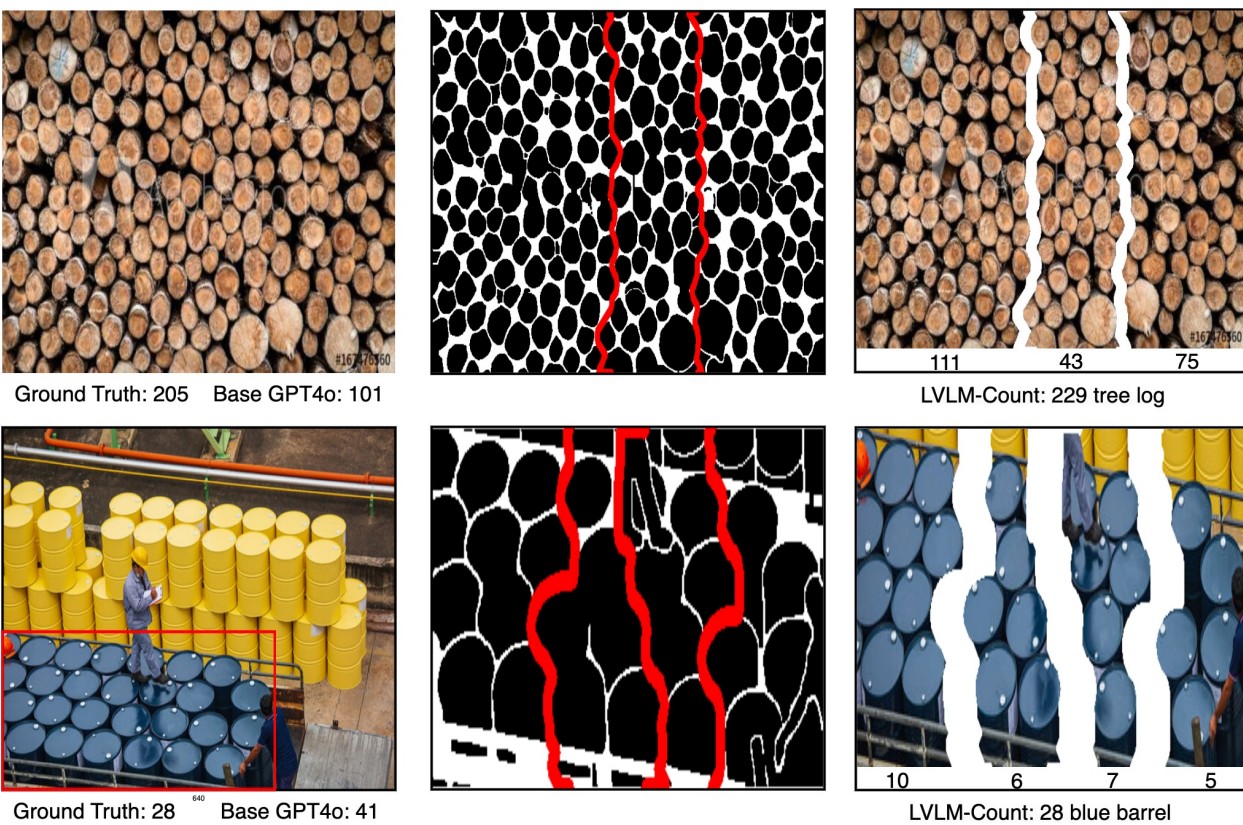

Figure 16: Performance of LVLM-Count on real-world applications in industry/warehousing. The top row shows an image of a stockpile of tree logs, with the objects of interest being "tree logs." The bottom row shows an aerial image of an industrial area containing barrels of various colors, with the objects of interest being "*blue* barrels."

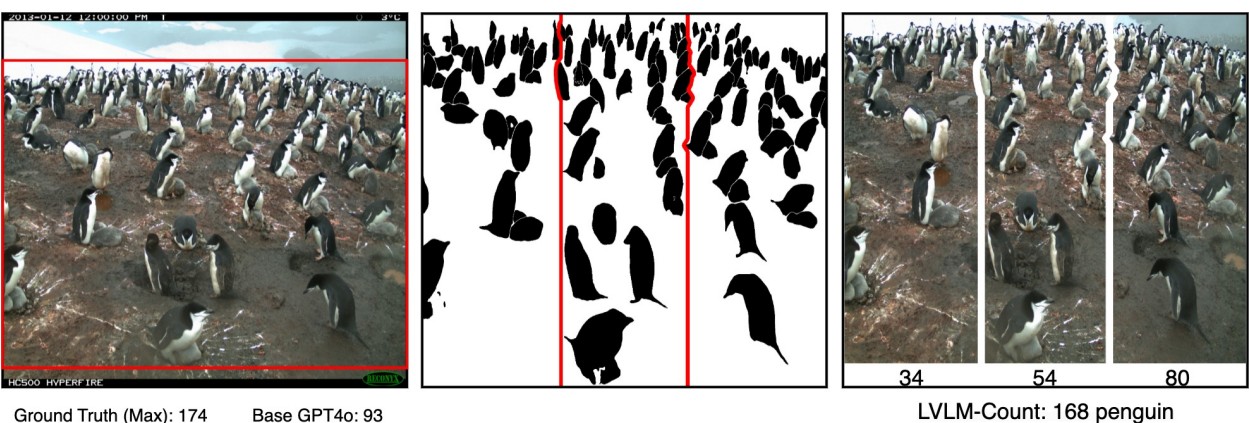

Figure 17: Performance of LVLM-Count on real-world applications in environmental monitoring. The image is sourced from (Penguin Research, 2016), an initiative to monitor the penguin population in Antarctica, with the objects of interest being "penguins."

counts the number of instances from both categories combined since the counting is ultimately performed by the LVLM. Note that the number of objects in this image is limited, and GPT-4o might answer these questions correctly without the need for the LVLM-Count pipeline. This image has been chosen to illustrate

LVLM-Count's power in handling multiple objects in a counting task rather than for comparison with the baseline LVLM.

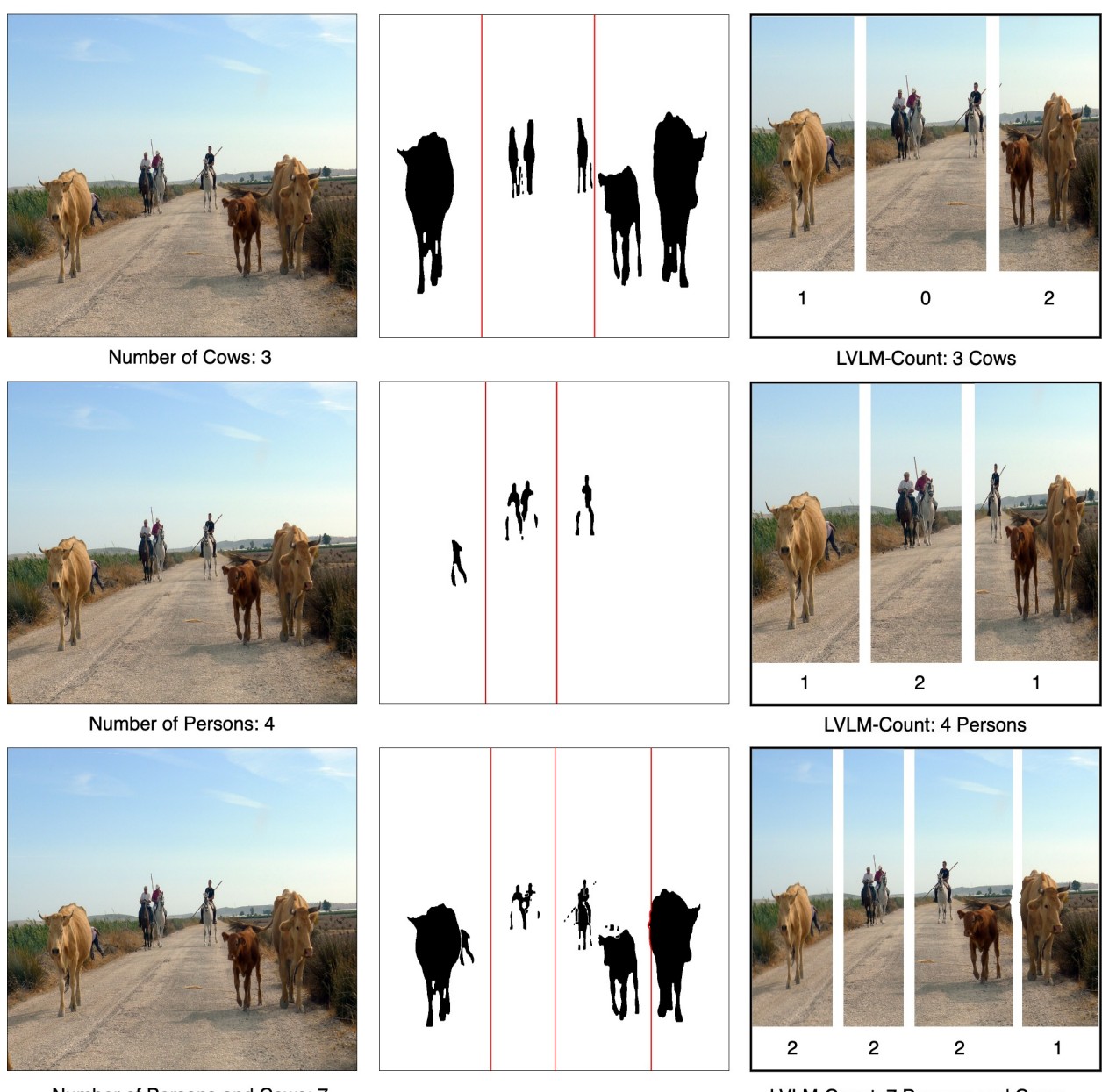

Figure 18: Illustration of the ability of LVLM-Count in counting an object of interest determined by a prompt when multiple object categories exist in a single image. Top row: Object of interest is "cow". Middle row: Object of interest is "person". Bottom row: Object of interest is "person and cow"

## G   False Positive Masks at the Target Segmentation Stage

One of the reasons we task an LVLM to count the objects in the subimages instead of using the number of generated masks at the target segmentation stage as the final count of the objects of interest is the existence of false positive masks. The GroundingDINO model is responsible for detecting the objects of interest, determined by expression $E$, and passing the output bounding boxes to SAM for producing segmentation

masks. Nonetheless, GroundingDINO is not as strong as an LVLM in understanding expressions extracted from complex questions. Thus, it often returns bounding boxes for all instances of the object category mentioned in the expression, even if those instances do not satisfy other conditions in the expression.

For example, in the top row of Figure 19, $E =$ "*brown* egg". However, all the eggs have been segmented regardless of their color. Thus, counting the masks results in a significant error. Interestingly, as we can see, the false positive masks do not negatively affect LVLM-Count's final answer, as the counting is done by an LVLM at the final stage, which is much stronger than GroundingDINO at understanding referring expressions. The only effect is that the white eggs have not been cut through by the division lines either. In the bottom row, we have chosen an image from the challenging Emoji-Count benchmark. The image contains icons, all of which have an arrow but point in different directions. However, the objects of interest are only "right arrows curving left." Similar to the eggs example, taking the masks used for object-aware division results in a significant error.

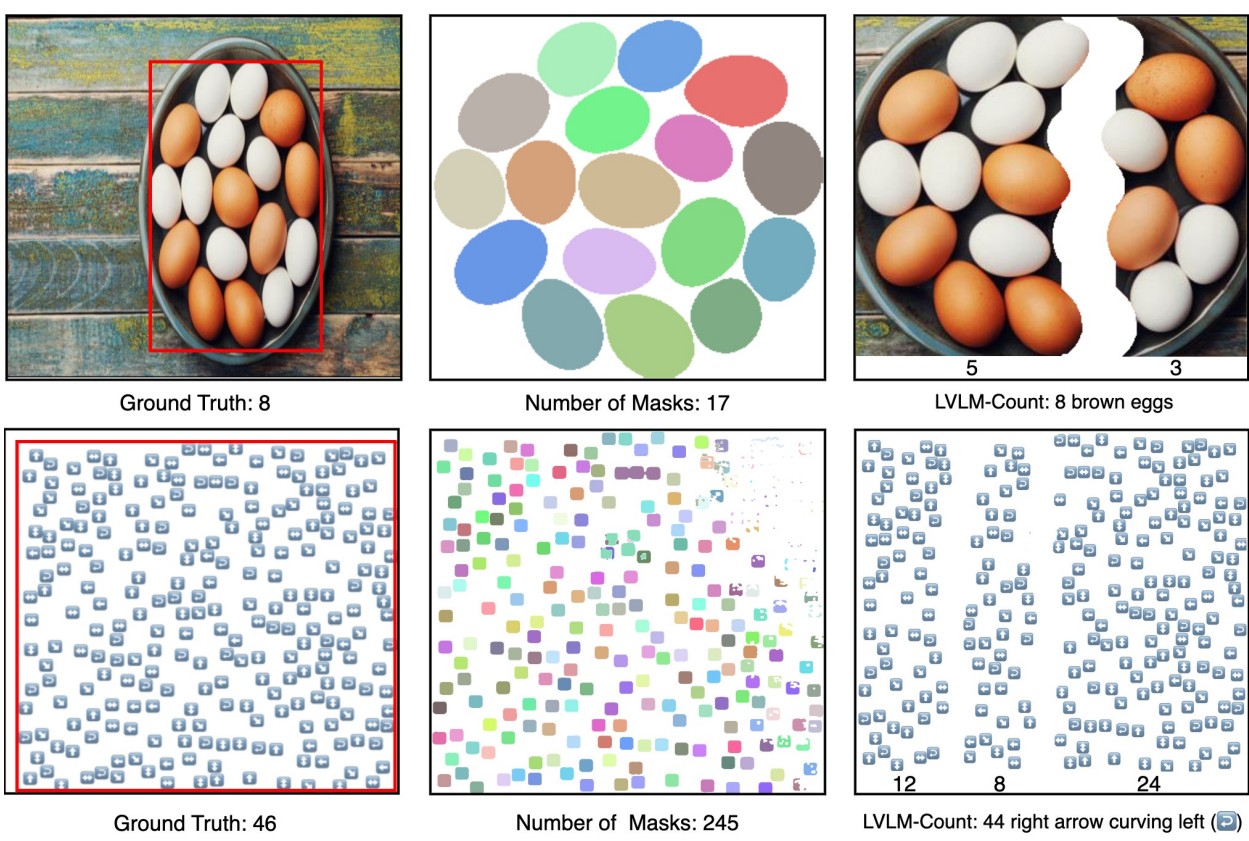

Figure 19: Top row: The object of interest is "*brown* egg." However, all the eggs have been segmented because of the limitation of the GroundingDINO model in understanding complex referring expressions. Regardless, LVLM-Count provides a significantly more accurate number compared to the number of masks. Bottom row: The object of interest is "right arrows curving left." Similar to the image of the eggs, counting the number of masks results in a very large error, while LVLM-Count provides a much more accurate number.

## H    Performance Analysis of LVLM-Count for different ground truth ranges on FSC-147 dataset

To further investigate the performance of our pipeline, we divide the ground truth values in the FSC-147 test set into intervals and plot the MAE for the base GPT-4o, Qwen2, and Gemma 3 models alongside the results from LVLM-Count using each model, as shown in Figure 20. The first interval contains relatively small ground truth values, a range in which LVLMs already perform well. As the ground truth values increase, the

base models exhibit increasingly larger errors compared to LVLM-Count, with the margin growing rapidly. This behavior is consistent with our observations of counting errors on the blue circles in Figure 2.

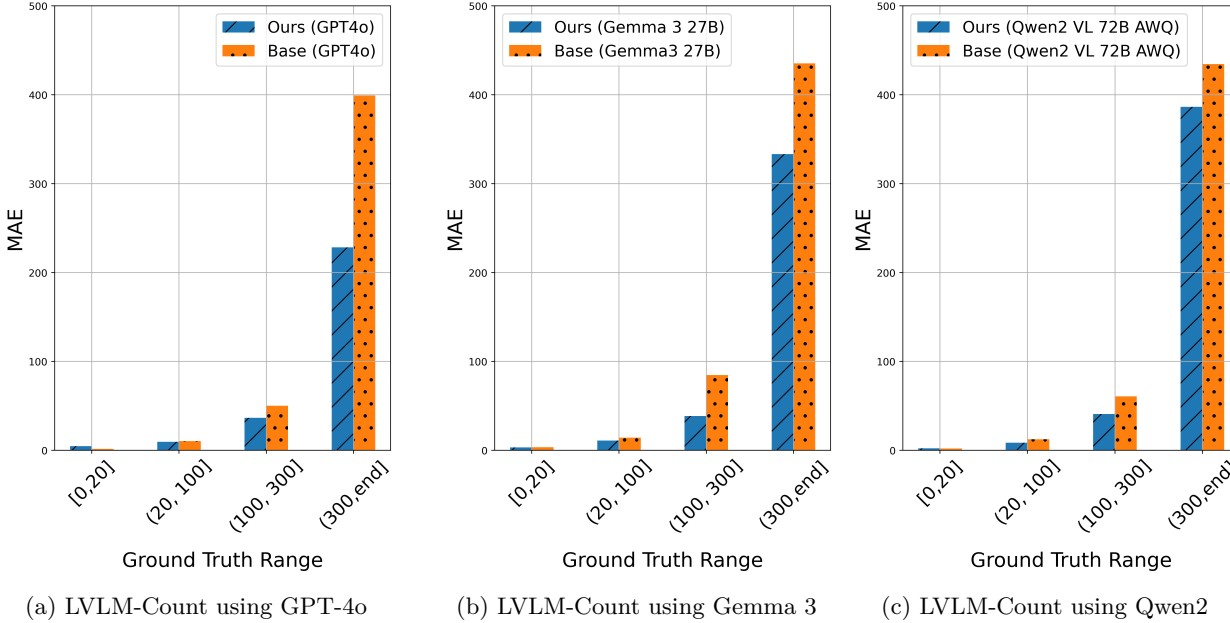

(a) LVLM-Count using GPT-4o  (b) LVLM-Count using Gemma 3  (c) LVLM-Count using Qwen2

Figure 20: Performance analysis of our method, LVLM-Count, on the FSC-147 test set using GPT-4o (Figure 20a), Gemma 3 (Figure 20b), and Qwen2 (Figure 20c). In the first interval, all base LVLMs exhibit a lower MAE. However, in intervals with higher ground truth values, LVLM-Count achieves a lower MAE compared to the base LVLMs. Note that as the ground truth values grow higher, the margin of improvement increases rapidly.

## I  Report of Various Accuracy Metrics for the Performance of LVLM-Count on the FSC-147 Dataset, PASCAL VOC, Emoji-Count, and Penguin Benchmarks

This section presents various accuracy measures for the experiments reported in Tables 1, 2, 3, and 4. The accuracy metrics are defined in Table 7. Following these definitions, Table 8 shows the measured accuracies for the FSC-147 test set. Similarly, Tables 9, 10, and 11 present the accuracy metrics for the PASCAL VOC, Emoji-Count, and Penguin benchmarks, respectively. The results in the tables demonstrate that LVLM-Count generally achieves a higher accuracy than the base LVLM it employs.

Table 7: Definitions of Various Accuracy Metrics. GT denotes the ground truth number.

| Metric | Definition |
|---|---|
| Acc | Percentage of answers such that $answer = GT$ |
| Acc$\pm$1 | Percentage of answers such that $|answer - GT| \leq 1$ |
| Acc$\pm$3 | Percentage of answers such that $|answer - GT| \leq 3$ |
| Acc$\pm$5 | Percentage of answers such that $|answer - GT| \leq 5$ |
| Acc$\pm$10 | Percentage of answers such that $|answer - GT| \leq 10$ |

## J  Inference Time Comparison

LVLM-Count is a pipeline consisting of multiple stages. However, in this section, we demonstrate that the dominant portion of the time during inference with the LVLMs tested in this paper, i.e. GPT4o, Gemma 3,

Table 8: FSC-147 Dataset. A (↑) next to the measured accuracies for LVLM-Count indicates improvement over the base LVLM it uses, while a (↓) indicates degradation compared to the corresponding base LVLM.

| Method | Acc (%) | Acc±1 (%) | Acc±3 (%) | Acc±5 (%) | Acc±10 (%) |
|---|---|---|---|---|---|
| GPT4o | 12.24 | 26.22 | 42.10 | 52.41 | 66.58 |
| LVLM-Count (GPT4o as LVLM) | 14.45 (↑) | 28.99 (↑) | 47.25 (↑) | 57.03 (↑) | 70.45 (↑) |
| Gemma 3 27B | 8.71 | 18.94 | 32.55 | 41.57 | 55.15 |
| LVLM-Count (Gemma 3 as LVLM) | 11.37 (↑) | 25.46 (↑) | 41.51 (↑) | 52.07 (↑) | 67.09 (↑) |
| Qwen2 VL 72B AWQ | 9.19 | 20.81 | 36.83 | 46.95 | 62.07 |
| LVLM-Count (Qwen2 VL 72B AWQ as LVLM) | 9.61 (↑) | 23.95 (↑) | 43.05 (↑) | 53.78 (↑) | 69.08 (↑) |

Table 9: PASCAL VOC Benchmark. A (↑) next to the measured accuracies for LVLM-Count indicates improvement over the base LVLM it uses, while a (↓) indicates degradation compared to the corresponding base LVLM.

| Method | Acc (%) | Acc±1 (%) | Acc±3 (%) | Acc±5 (%) | Acc±10 (%) |
|---|---|---|---|---|---|
| GPT4o | 30.39 | 47.06 | 61.11 | 71.90 | 89.87 |
| LVLM-Count (GPT4o as LVLM) | 32.35 (↑) | 51.96 (↑) | 73.20 (↑) | 82.35 (↑) | 93.46 (↑) |
| Gemma 3 27B | 29.41 | 54.58 | 75.49 | 85.62 | 92.16 |
| LVLM-Count (Gemma 3 27B as LVLM) | 30.39 (↑) | 50.98 (↓) | 78.43 (↑) | 85.62 (-) | 94.77 (↑) |
| Qwen2 VL 72B AWQ | 24.18 | 42.48 | 60.13 | 72.22 | 86.60 |
| LVLM-Count (Qwen2 VL 72B AWQ as LVLM) | 28.76 (↑) | 46.73 (↑) | 67.32 (↑) | 78.43 (↑) | 89.87 (↑) |

Table 10: Emoji-Count Benchmark. A (↑) next to the measured accuracies for LVLM-Count indicates improvement over the base LVLM it uses, while a (↓) indicates degradation compared to the corresponding base LVLM.

| Method | Acc (%) | Acc±1 (%) | Acc±3 (%) | Acc±5 (%) | Acc±10 (%) |
|---|---|---|---|---|---|
| GPT4o | 1.85 | 5.54 | 13.73 | 21.12 | 43.94 |
| LVLM-Count (GPT4o as LVLM) | 2.81 (↑) | 9.88 (↑) | 23.53 (↑) | 36.22 (↑) | 60.24 (↑) |
| Gemma 3 27B | 0.80 | 1.77 | 4.10 | 7.07 | 16.14 |
| LVLM-Count (Gemma 3 27B as LVLM) | 2.49 (↑) | 5.30 (↑) | 13.33 (↑) | 19.92 (↑) | 38.63 (↑) |
| Qwen2 VL 72B AWQ | 0.88 | 2.89 | 7.55 | 11.41 | 19.76 |
| LVLM-Count (Qwen2 VL 72B AWQ as LVLM) | 2.33 (↑) | 6.75 (↑) | 16.22 (↑) | 25.70 (↑) | 43.86 (↑) |

Table 11: Penguin Benchmark. A (↑) next to the measured accuracies for LVLM-Count indicates improvement over the base LVLM it uses, while a (↓) indicates degradation compared to the corresponding base LVLM.

| Method | Acc (%) | Acc±1 (%) | Acc±3 (%) | Acc±5 (%) | Acc±10 (%) |
|---|---|---|---|---|---|
| GPT4o | 2.00 | 3.67 | 8.33 | 12.00 | 22.00 |
| LVLM-Count (GPT4o as LVLM) | 2.67 (↑) | 6.33 (↑) | 14.33 (↑) | 18.67 (↑) | 33.00 (↑) |
| Gemma 3 27B | 1.33 | 1.67 | 7.00 | 8.67 | 16.00 |
| LVLM-Count (Gemma 3 27B as LVLM) | 1.00 (↓) | 3.67 (↑) | 9.33 (↑) | 12.67 (↑) | 23.67 (↑) |
| Qwen2 VL 72B AWQ | 1.67 | 2.00 | 6.33 | 10.33 | 16.67 |
| LVLM-Count (Qwen2 VL 72B AWQ as LVLM) | 1.67 (-) | 5.33 (↑) | 11.67 (↑) | 18.00 (↑) | 31.00 (↑) |

and Qwen2, is spent querying the LVLM to count objects, and the additional steps in the pipeline comprise a small portion in comparison.

In the four stages of the pipeline, the following main steps are taken: making a call to an LLM/LVLM to extract the object of interest from $Q$, calling GroundingDINO for area detection, calling GroundingDINO and SAM for target segmentation, performing some post-processing on the masks, running the mean-shift clustering algorithm, running the $A^*$ search, and querying the LVLM for the sub-images. From the listed

steps, the running time of the post-processing on masks is negligible. The time required for the rest of the steps in an optimized pipeline is demonstrated in Table 12. To obtain the inference time for each step, we evaluated it on 1,000 images from the FSC-147 dataset and calculated the average.

Table 12: Inference time breakdown of the LVLM-Count pipeline for different LVLMs (Time in seconds)

| Method | Extract $E$ from $Q$ | Area Detection | Target Segmentation | Clustering | $A^*$ | Subimage Analysis[3] | Total |
|---|---|---|---|---|---|---|---|
| LVLM-Count (GPT4o) | 0.11 | | | | | 1.65 | 2.56 |
| LVLM-Count (Gemma 3 27B) | 0.11 | 0.10 | 0.43 | 0.24 | 0.03 | 1.11 | 2.02 |
| LVLM-Count (Qwen2 VL 72B AWQ) | 0.10 | | | | | 1.17 | 2.07 |

As shown in Table 12, the dominant term in the inference time comes from calling the LVLM to count objects of interest in the sub-images. This occupies more than half of the total inference time. However, a similar step occurs when querying the base LVLM to count objects of interest in the main image. The additional steps present in LVLM-Count but absent in the base LVLM constitute a smaller portion of the time. Table 13 presents a comparison between the total inference time of LVLM-Count and the base LVLMs. This increase in inference time may be acceptable for most applications, given the substantial improvements in counting accuracy when using LVLM-Count.

Table 13: Comparison of the inference time between the base LVLM and LVLM-Count

| Method | Inference (s) |
|---|---|
| Base GPT4o | 1.68 |
| LVLM-Count (Using GPT-4o) | 2.56 |
| Base Gemma 3 27B | 1.52 |
| LVLM-Count (Using Gemma 3 27B) | 2.01 |
| Base Qwen2 VL 72B AWQ | 1.46 |
| LVLM-Count (Using Qwen2 VL 72B AWQ) | 2.07 |

# K Comparison of Base LVLMs and LVLM-Count's Counting Performance with SOTA Counting Models as a Reference Point

LVLM-Count is designed to enhance the counting performance of LVLMs. Tables 1, 2, 3, and 4 in the main paper present extensive experimental results demonstrating LVLM-Count's accuracy improvements over the base LVLMs. In this section, we include the results of state-of-the-art (SOTA) dedicated counting models for the experiments reported in the main paper. Specifically, we compare the performance of GROUNDINGREC Dai et al. (2024), COUNTGD Amini-Naieni et al. (2024), and DAVE$_{prm}$ Pelhan et al. (2024), as well as TFOC Shi et al. (2024). The first three models have counting-specific training, while the latter, though a dedicated counting model, does not require counting-specific training. Reporting these results provides additional insights into the progress of counting performance in LVLMs and their enhanced performance with LVLM-Count.

Table 14 presents the results of dedicated counting models alongside LVLMs and their corresponding LVLM-Count variants on the FSC-147 test set. Note that GroundingREC, CountGD, and DAVE$_{prm}$ are the top three highest-performing text-based trained models on FSC-147 in the literature. Moreover, TFOC is the best-performing training-free model on FSC-147. We observe that, in general, the dedicated counting models (henceforth referred to simply as counting models) outperform the base LVLMs. However, the use of LVLM-Count significantly narrows this performance gap.

Results on the PASCAL VOC dataset are reported in Table 15. For training-based counting models, this is considered a cross-dataset evaluation, meaning that they are trained on the training set of FSC-147 and tested on the PASCAL VOC benchmark. We observe that some trained counting models still outperform

---

[3]This is for a case where sub-image queries to the LVLM are made in parallel

Table 14: Evaluation of SOTA counting models on the FSC-147 test set serves as a reference point for assessing the performance of LVLMs and their corresponding LVLM-Count variants. The column "Trained Model" indicates whether a model has been trained on FSC-147. Columns marked with $\Delta$ show the performance difference between LVLM-Count and its base LVLM. Improvements are shown in green, while degradations are shown in red.

| Method | Trained Model | MAE ↓ | $\Delta$ | RMSE ↓ | $\Delta$ |
|---|---|---|---|---|---|
| TFOC (Shi et al., 2024) | ✗ | 24.79 | - | 137.15 | - |
| DAVE$_{prm}$ (Pelhan et al., 2024) | ✓ | 14.90 | - | 103.42 | - |
| CountGD (Amini-Naieni et al., 2024) | ✓ | 14.76 | - | 120.42 | - |
| GroundingREC (Dai et al., 2024) | ✓ | **10.12** | - | 107.19 | - |
| GPT-4o | ✗ | 25.57 | - | 137.26 | - |
| LVLM-Count (GPT-4o as LVLM) | ✗ | 17.86 | ↓ 7.71 | **91.71** | ↓ 45.55 |
| Gemma 3 27B | ✗ | 30.59 | - | 132.61 | - |
| LVLM-Count (Gemma 3 27B as LVLM) | ✗ | 20.25 | ↓ 10.34 | 110.45 | ↓ 22.16 |
| Qwen2 VL 72B AWQ | ✗ | 34.48 | - | 149.49 | - |
| LVLM-Count (Qwen2 VL 72B AWQ as LVLM) | ✗ | 22.29 | ↓ 11.89 | 119.46 | ↓ 30.44 |

LVLMs; however, this is likely due to the significant overlap between the categories in PASCAL VOC and FSC-147. Nevertheless, LVLM-Count is very effective in narrowing the performance gap. Additionally, the results on the Emoji-Count benchmark are reported in Table 16. This is also a cross-dataset evaluation. In contrast to PASCAL VOC, Emoji-Count consists of complex concepts and categories that do not overlap with FSC-147. As a result, LVLMs, which have far stronger generalization power for unseen data and complex concepts, outperform counting models by a large margin. Employing LVLM-Count widens this gap even further.

Table 15: Evaluation of SOTA counting models on the PASCAL VOC counting benchmark as a reference point for the performance of LVLMs and their corresponding LVLM-Count performance.

| Method | MAE ↓ | $\Delta$ | RMSE ↓ | $\Delta$ |
|---|---|---|---|---|
| TrainingFree (Shi et al., 2024) | 12.03 | - | 18.18 | - |
| DAVE$_{prm}$ (Pelhan et al., 2024) | 12.39 | - | 22.81 | - |
| CountGD (Amini-Naieni et al., 2024) | **2.81** | - | 7.01 | - |
| GroundingRec (Dai et al., 2024) | 4.05 | - | 7.80 | - |
| GPT4o | 4.64 | - | 8.56 | - |
| LVLM-Count (GPT4o as LVLM) | 3.42 | ↓ 1.22 | 7.17 | ↓ 1.39 |
| Gemma 3 27B | 3.40 | - | 7.54 | - |
| LVLM-Count (Gemma 3 27B as LVLM) | 2.97 | ↓ 0.0.43 | **6.16** | ↓ 1.38 |
| Qwen2 VL 72B AWQ | 4.80 | - | 8.71 | - |
| LVLM-Count (Qwen2 VL 72B AWQ as LVLM) | 4.12 | ↓ 0.68 | 7.76 | ↓ 0.95 |

Finally, Table 17 presents the results of prior counting models on the Penguin benchmark alongside the counting performance of LVLMs. We observe that prior training-based counting models significantly outperform LVLMs on this benchmark. However, similar to other datasets, LVLM-Count narrows this performance gap.

## L  LVLM-Count's Ability to Handle Visual Exemplars

Exemplar-based counting methods face challenges due to the difficulty of acquiring representative exemplars. Additionally, they often struggle with intra-class variability, as objects within the same category may exhibit diverse appearances, leading to noisy matches and reduced accuracy. In contrast, text-based methods offer

Table 16: Evaluation of SOTA counting models on the Emoji-Count benchmark, serving as a reference point for the performance of LVLMs and their corresponding LVLM-Count performance.

| Method | MAE ↓ | Δ | RMSE ↓ | Δ |
|---|---|---|---|---|
| TFOC (Shi et al., 2024) | 64.64 | - | 87.45 | - |
| DAVE$_{prm}$ (Pelhan et al., 2024) | 198.99 | - | 208.08 | - |
| CountGD (Amini-Naieni et al., 2024) | 137.93 | - | 156.80 | - |
| GroundingREC (Dai et al., 2024) | 143.22 | - | 158.74 | - |
| GPT-4o | 23.57 | - | 36.97 | - |
| LVLM-Count (GPT-4o as LVLM) | 16.57 | ↓ 7 | 33.11 | ↓ 3.86 |
| Gemma 3 27B | 21.39 | - | 24.04 | - |
| LVLM-Count (Gemma3 27B as LVLM) | **16.16** | ↓ 5.23 | **21.27** | ↓ 2.77 |
| Qwen2 VL 72B AWQ | 78.05 | - | 159.22 | - |
| LVLM-Count (Qwen2 VL 72B AWQ as LVLM) | 24.43 | ↓ 53.62 | 43.38 | ↓ 115.84 |

Table 17: Evaluation of SOTA counting models on the Penguin benchmark, serving as a reference point for the performance of LVLMs and their corresponding LVLM-Count performance.

| Method | MAE ↓ | Δ | RMSE ↓ | Δ |
|---|---|---|---|---|
| TFOC (Shi et al., 2024) | 59.34 | - | 74.40 | - |
| DAVE$_{prm}$ (Pelhan et al., 2024) | 22.29 | - | 31.38 | - |
| CountGD (Amini-Naieni et al., 2024) | **17.07** | - | **25.08** | - |
| GroundingREC (Dai et al., 2024) | 22.04 | - | 26.75 | - |
| GPT4o | 35.18 | - | 45.76 | - |
| LVLM-Count (GPT-4o as LVLM) | 26.76 | ↓ 8.42 | 38.60 | ↓ 7.16 |
| Gemma 3 27B | 49.44 | - | 60.20 | - |
| LVLM-Count (Gemma 3 27B as LVLM) | 34.13 | ↓ 15.31 | 44.11 | ↓ 16.09 |
| Qwen2 VL 72B AWQ | 44.02 | - | 65.59 | - |
| LVLM-Count (Qwen2 VL 72B AWQ as LVLM) | 28.21 | ↓ 15.81 | 40.40 | ↓ 25.19 |

greater flexibility, as textual descriptions are easily provided, modifiable, and capable of encoding abstract or fine-grained concepts. This adaptability makes text-based approaches particularly suitable for dynamic or open-set scenarios, where predefined exemplars are impractical.

Our focus in this work, similar to many recent works Liu et al. (2023); Amini-Naieni et al. (2023), is on text-based counting due to the above-mentioned reasons. Nevertheless, LVLM-Count is capable of working with visual exemplars alone as well as combinations of text prompts and visual exemplars, in addition to its text-only capabilities. The adaptation is straightforward, as most LVLMs also accept visual prompts. In Table 18, we evaluate the base GPT-4o and LVLM-Count (which uses GPT-4o) on Emoji-Count with text-only prompts, visual exemplars, and combinations of text and visual exemplars. For the visual-exemplar version, a single image of the target emoji icon is provided, and the LVLM is required to count similar instances. For the combined text-and-visual version, an image of the target emoji icon alongside its name is provided to the model. The results show higher accuracy when combining text and visual exemplars, while the text-only and exemplar-only versions achieve approximately the same performance.

## M   Chain of Thought Prompting for LVLM-Count

In this work, our primary focus is on developing a pipeline to enhance the counting ability of LVLMs for counting prompts in general, rather than optimizing for a specific prompting method. Nevertheless, we investigated the impact of Chain-of-Thought (CoT) prompting Wei et al. (2022) on visual counting task. For our experiments, we appended the phrase "think step-by-step" to the counting prompt. As shown in Table 19,

Table 18: Evaluation of LVLM-Count's performance using text-only inputs, exemplar-only inputs, and a combination of text and exemplars on the Emoji-Count dataset.

| Method | MAE ↓ | Δ | RMSE ↓ | Δ |
|---|---|---|---|---|
| GPT4o | 22.51 | - | 35.94 | - |
| LVLM-Count (GPT4o as LVLM, text) | 16.29 | ↓ 6.22 | 32.47 | ↓ 3.47 |
| LVLM-Count (GPT4o as LVLM, visual exemplars) | 15.66 | ↓ 6.85 | 31.69 | ↓ 4.25 |
| LVLM-Count (GPT4o as LVLM, text + visual exemplars) | 13.46 | ↓ 9.05 | 24.35 | ↓ 11.59 |

the results on the PASCAL VOC benchmark reveal that CoT negatively affects counting performance—not only for LVLM-Count but also for the base LVLM.

Table 19: Evaluation of CoT prompting for visual counting on PASCAL VOC.

| Method | MAE ↓ | RMSE ↓ |
|---|---|---|
| GPT4o | 4.46 | 8.35 |
| GPT4o with CoT prompting | 5.69 | 10.10 |
| LVLM-Count (GPT4o as LVLM) | 3.31 | 7.11 |
| LVLM-Count with CoT prompting (GPT4o as LVLM) | 3.89 | 7.70 |

# N    Illustration of the Complete Workflow of the LVLM-Count for an Additional Image

In this section, we demonstrate the same steps illustrated for the example image of eggs in Figures 3, 4, 5, and 6 for an image of zebras drinking water.

The zebra image is passed to the pipeline along with the question $Q$ ="how many zebras are in the image?". First, $E$ ="zebra" is extracted using the LLM. Then the zebra image is passed to the area detection stage, where the prompt given to GroundingDINO is "zebras". The output bounding boxes are merged, and the resulting area is cropped, as illustrated in Figure 21. The cropped area is then passed to the target segmentation stage. At this stage, GroundingDINO detects the objects of interest defined by $E$ as the input prompt. SAM then uses the output bounding boxes to produce segmentation masks for the zebras, as shown in Figure 22.

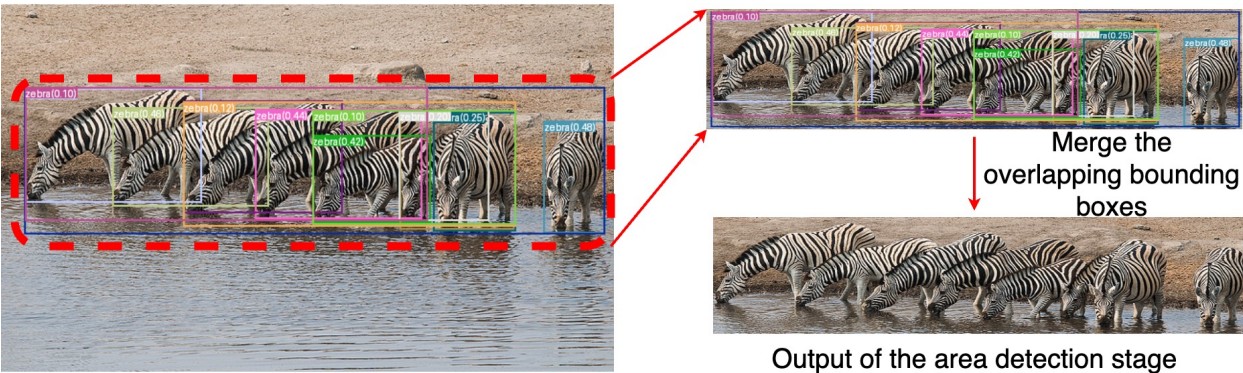

Figure 21: Illustration of the area detection step of LVLM-Count for the zebra image. For this image, $Q$ is set to "How many zebras are in the image?". The LLM used in this step returns an $E$, which is "zebra". The plural form of $E$, "zebras", and the original image are given as input to GroundingDINO, which returns some bounding boxes (left and upper right images) that are merged to form the final detected area.

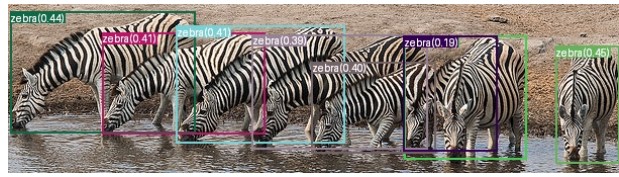 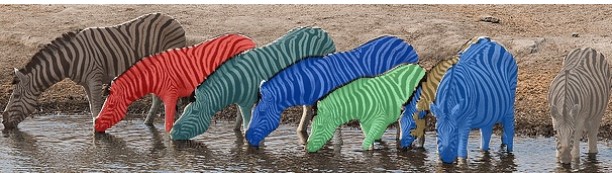

(a) GroundingDINO output  (b) SAM output

Figure 22: Illustration of the target segmentation step of LVLM-Count for the zebra image. The goal is to produce all the instance masks for $E$ set to "zebra". The cropped image from Figure 21, together with $E$, is given as input to GroundingDINO, which produces the output shown in Figure 22a. Figure 22a is then given as input to SAM, which produces the output shown in Figure 22b.

After the target segmentation stage, the masks are passed to the object-aware division stage. First, the masks are used in the cluster-based approach to find the location of the start and end points of the division paths, i.e., $(P_s^1, P_e^1)$ and $(P_s^2, P_e^2)$. Then these masks are turned into a black-and-white image, which, in turn, is mapped to a graph. The division paths are then found by connecting each start point to its corresponding end point by running the $A^*$ search algorithm on the graph. The found paths are mapped back into the image domain and drawn in red, as depicted in Figure 23. The image contours are determined based on the drawn red paths, and each contour's interior is masked out independently to obtain the subimages. Finally, the subimages are given to the LVLM to count the number of zebras in each.

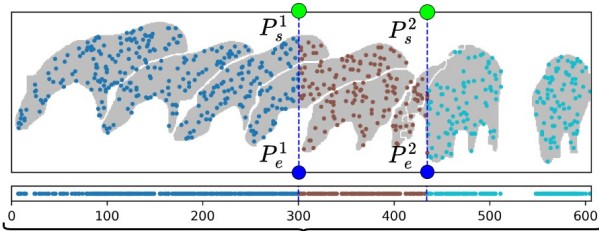 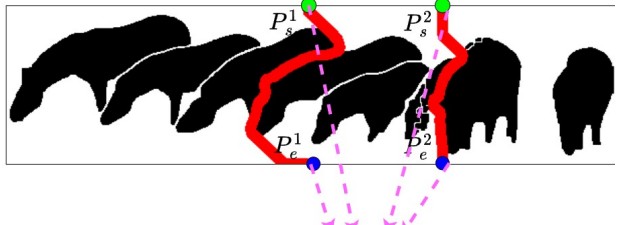

Projection of samples onto the x-axis  End points found by the cluster-based aproach

Figure 23: Left: Illustration of the unsupervised and non-parametric method to obtain the division points $(P_s^1, P_e^1)$ and $(P_s^2, P_e^2)$. First, a few pixels are sampled (shown as points inside the segmented objects) from the pixels composing each mask. The samples are projected onto the $x$-axis. The projected points are clustered using mean-shift clustering. The point in the middle of two consecutive clusters is considered a vertical division point. The straight vertical lines are drawn just for better visualization of the division points. Right: Illustration of object-aware division. The masks from Figure 22b are turned into a black-and-white image. A dividing path is found by connecting $P_s$ to $P_e$ using the $A^*$ search algorithm in a graph that corresponds to the binary image, where the only nodes in the graph are white pixels, which are connected to all of their white neighboring pixels. The path is mapped back to the pixel domain.

## O  Emoji-Count Benchmark Details

In this section, we provide further details about the Emoji-Count benchmark. The full benchmark, including all images and corresponding annotations, is publicly available in our GitHub repository. The emoji icons used in daily chats can be placed in classes closely related to a specific topic. While the icons within these groups often appear very similar, they possess subtle yet important differences. Identifying a specific emoji among others based on these fine distinctions requires a complex level of understanding from the counting model.

Our dataset was built from the 1,816 standard Apple emojis available at the time of its creation. Each emoji has a unique descriptive name. We first grouped these emojis into 82 non-overlapping classes based on their name's topic. To prevent ambiguity for a counting model, we then removed confusing icons, such as those with names that could also refer to a more specific emoji in the class. This cleaning process left us with 1,197

icons. From each class, we randomly selected six icons for inclusion; classes with fewer than six retained all their icons. Figure 25b shows the six selected icons from two example classes.

To generate the benchmark images, we began with an empty $1024 \times 1024$ pixel white canvas for each class. We then populated each canvas with instances of the icons from that class. Note that each icon may also be referred to as an object category. The procedure was as follows: for each of the six icons in a class, we generated a random number between 30 and 50. A number of instances equal to this random value were then placed at random positions on the canvas, ensuring that the icons did not overlap and remained completely within the canvas borders. Each image is annotated with the count for every icon type it contains. The benchmark comprises 415 questions in total. Table 20 provides additional statistics for the benchmark, while Figure 25a shows the distributions of the ground truth counts.

Table 20: Statistics of the Emoji-Count benchmark dataset.

| Metric | Value |
|---|---|
| Total Questions | 415 |
| Total Classes | 82 |
| Total Images | 82 |
| Total Categories | 415 |
| Average Questions per Image | 5.06 |
| Answer Range | 30-50 |
| Average Answer Value | $39.87 \pm 6.33$ |

## P    Parsing Strategy for the Output of Open-source LVLMs

Unlike GPT-4o, which provides the user with the ability to format the output answer using a JSON schema, the open-source models used in this work, i.e., Qwen2 VL 72B AWQ and Gemma 3 27B, do not have such an option. When we prompt these models to count the number of instances of an object of interest, the answer is often accompanied by descriptive text that does not follow a specific format, making the extraction of the predicted object count difficult. To solve this issue, we include a sentence in the counting prompt given to the LVLM that asks the model to put its final prediction inside double brackets. The template prompt is: "How many `object-of-interest` are in the image? Report your final answer with only one number inside [[Double Brackets]]".

After the model generates an answer, we use Python's regular expression package to extract the predicted number from within the double brackets. Often, this number is a word rather than a numeral. Therefore, after extraction, we use Python's `text2num` package to convert the word into its numerical form. The following code snippet shows this number extraction process. Details unrelated to the regular expression have been omitted.

```python
import re
from text_to_num import alpha2digit

text_prompt = ("How many object-of-interest are in the image. "
               "Report your final answer with only one number inside [[Double Brackets]]")
while True:
    try:
        output_text = LVLM( text_prompt, image)
        answer = re.search("\[\[(.+)\]\]", output_text[0]).group(1)
        count = alpha2digit(answer, "en")
        count = int(count)
        break
```

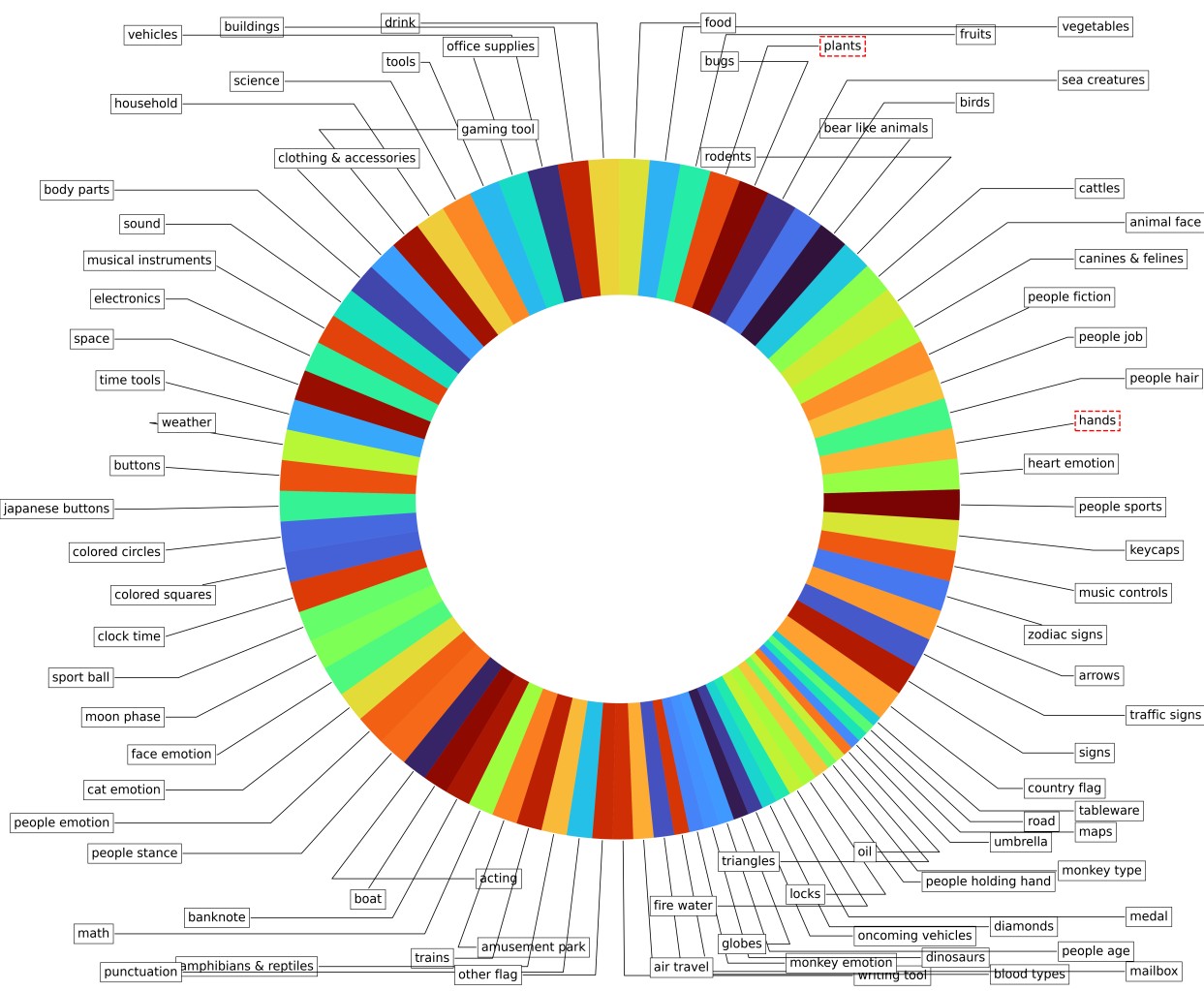

Figure 24: This pie chart illustrates the 82 classes in the Emoji-Count benchmark. Inside each class there are 3 to 6 type of emoji icons that are related to the topic of the class. For example, the emoji icons inside the plants and hands classes, have been demonstrated in Figure 25b

```
13      except Exception:
14          pass
```

Despite the format instruction in the prompt, the models might occasionally fail to produce an answer in the required format. This results in an error, which is handled by the code above. The model is then asked to generate a new output until a successful prediction is extracted.

Let us refer to an unsuccessful attempt to extract the predicted number as a failure. We measure the average failure rate per image and report it for each of the datasets in the main paper for the two open-source models that we use, i.e., Qwen2 VL 72B AWQ and Gemma 3 27B. These results are reported in Table 21.

## Q  Non-zero Prediction of LVLMs for Subimages with No Objects of Interest

As mentioned, LVLMs occasionally predict non-zero values for images that contain no object of interest. In our pipeline, it is very rare for subimages generated from images that contain a non-zero number of objects of interest to not contain any objects of interest. The area detection stage ensures that areas containing objects of interest are cropped, and the clustering step in the object-aware division stage finds the division points so

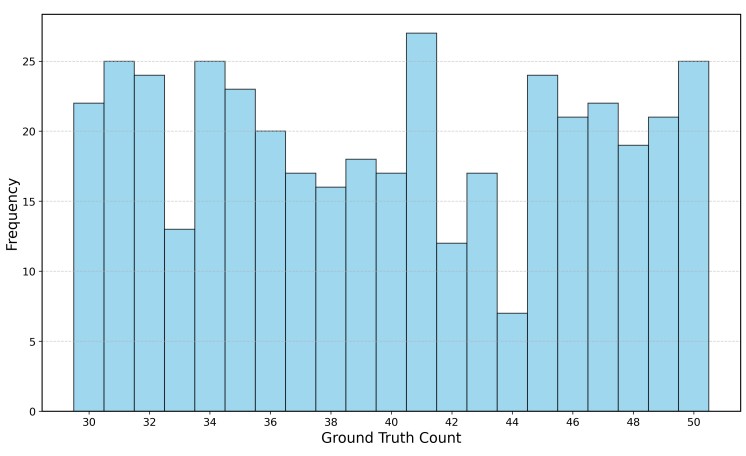

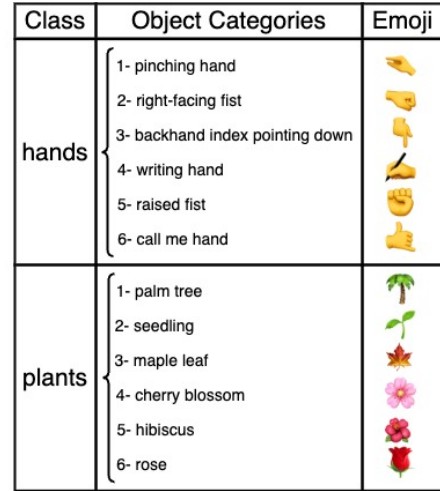

(a) Emoji-Count ground truth count distribution

(b) Two class examples from Figure 24

Figure 25: The figure illustrates: (a) the distribution of ground truth counts in the images of Emoji-Count benchmark, and (b) the object categories of two example classes, plants and hands

Table 21: Average failure rate of extracting the predicted number of open-source models across different benchmarks

| Method | Failure Rate | | | |
|---|---|---|---|---|
| | FSC-147 | PASCAL VOC | Emoji-Count | Penguin |
| Base Gemma 3 27B | $\approx 0$ | 0 | $\approx 0$ | $\approx 0$ |
| LVLM-Count (Using Gemma 3 27B) | $\approx 0$ | 0 | $\approx 0$ | 0 |
| Base Qwen2 VL 72B AWQ | 0.07 | 0.06 | 0.09 | 0.13 |
| LVLM-Count (Using Qwen2 VL 72B AWQ) | 0.07 | 0.08 | 0.01 | 0.12 |

that the distribution of potential objects of interest in different subimages is balanced. However, we provide quantitative information on this phenomenon for our PASCAL VOC benchmark.

In total, for only 9.80% of the images in the benchmark, at least one of the subimages generated through our pipeline contains no instance of the queried objects. Half of these cases are images that have a zero ground truth count to begin with. Excluding those cases, the percentage drops to 4.90%. We report the average non-zero prediction rate for the subimages with a zero ground truth count and the average predicted values in Table 22. For each LVLM, we report two cases. In one case, the input text is simply a counting prompt; in the other case, the input text includes the following sentence in addition to the counting prompt: "Say 0 if you do not see any." For the sake of brevity in the table entries, let us refer to this simple technique as Zero-aware prompt. As observed in the table, this simple technique is quite effective in reducing the error. Note that the standard LVLM-Count uses this technique.

Table 22: Average non-zero prediction rate and average predicted values for the subimages with zero ground truth count in PASCAL VOC benchmark.

| Method | Average Non-zero Prediction Rate ↓ | Average Predicted Value ↓ |
|---|---|---|
| LVLM-Count (GPT4o w/o Zero-aware prompt) | 0.39 | 0.78 |
| LVLM-Count (GPT4o) | 0.15 | 0.15 |
| LVLM-Count (Gemma 3 27B w/o Zero-aware prompt) | 0.45 | 2.24 |
| LVLM-Count (Gemma 3 27B) | 0.18 | 0.69 |
| LVLM-Count (Qwen2 VL 72B AWQ w/o Zero-aware prompt) | 0.42 | 0.51 |
| LVLM-Count (Qwen2 VL 72B AWQ) | 0.24 | 0.27 |

# R   Reproducibility Table for Implementation Parameter

This section provides Table 23, which lists the values of key parameters used in our implementation.

- **Detection threshold for area detection:** This threshold controls the acceptance of bounding boxes proposed by GroundingDINO, based on the probability of each box during the area detection stage.

- **Detection threshold for target segmentation:** This parameter is defined similarly but applies to the target segmentation stage.

- **Mask NMS IoU threshold:** This sets the maximum acceptable Intersection over Union (IoU) value between two masks in the NMS algorithm. If two masks exceed this IoU, the one with the lower probability is removed.

- **Mask erosion thickness:** This determines the size of a square kernel (`cv2.erode`) used to erode the outer layer of masks. This prevents obstructions for division paths when two adjacent masks overlap.

- **Mask refinement thickness:** This defines the size of a square kernel (`cv2.morphologyEx`) used to refine the surface of potentially pitted masks.

Table 23: Reproducibility table for implementation parameters

| Parameter | Value |
|---|---|
| Detection threshold for area detection | 0.1 |
| Detection threshold for target segmentation | 0.1 |
| Mask NMS IoU threshold | 0.4 |
| Mask erosion thickness | 2 pixels |
| Mask refinement thickness | 3 pixels |

# S   Dividing Images Along Both the Horizontal and Vertical Axes

As described in Section 3.3, in the object-aware division stage of LVLM-Count, the divisions are vertical. This is to keep the pipeline simple. Nonetheless, it is possible to perform the same steps to obtain horizontal division paths. In Table 24, we report the performance of a variant of LVLM-Count that leverages horizontal divisions in addition to vertical divisions on the FSC-147 dataset. Additionally, Figure 26 shows a visual example of this variant's performance on a sample from the FSC-147 dataset.

Table 24: Evaluation of a version of LVLM-Count that leverages divisions along both vertical and horizontal axes on the FSC-147 dataset.

| Method | MAE ↓ | Δ | RMSE ↓ | Δ |
|---|---|---|---|---|
| GPT-4o | 24.48 | - | 125.38 | - |
| LVLM-Count (GPT-4o as LVLM, vertical & horizontal divisions) | 17.67 | ↓ 6.81 | 90.61 | ↓ 34.47 |
| Gemma 3 27B | 30.59 | - | 132.61 | - |
| LVLM-Count (Gemma 3 27B as LVLM, vertical & horizontal divisions) | 18.44 | ↓ 12.15 | 103.91 | ↓ 28.7 |
| Qwen2 VL 72B AWQ | 34.18 | - | 149.49 | - |
| LVLM-Count (Qwen2 VL 72B AWQ as LVLM, vertical & horizontal divisions) | 18.19 | ↓ 15.99 | 104.12 | ↓ 45.37 |

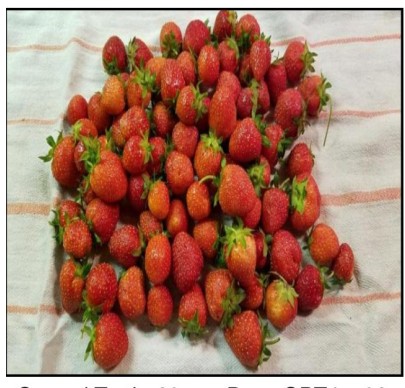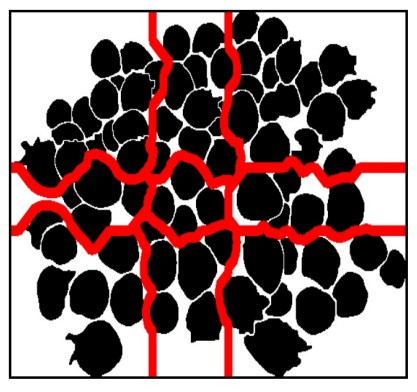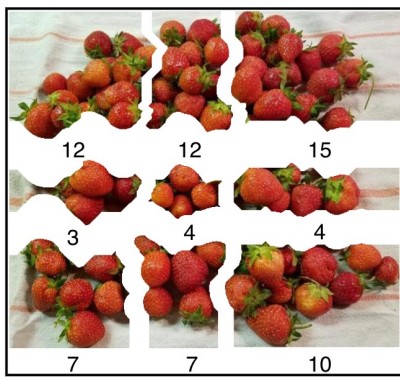

Ground Truth: 69     Base GPT4o: 96              LVLM-Count: 74 strawberries

Figure 26: A visual example of the performance of LVLM-Count on a sample from FSC-147 when it leverages both horizontal and vertical divisions

## T   Evaluation of the counting methods on TallyQA-Complex

TallyQA (Acharya et al., 2019) is an open-world counting dataset that includes complex counting questions involving relationships between objects, attribute identification, reasoning, and more. It is a fairly large dataset, with the training set containing 249318 questions and the test set having 22991 simple and 22991 complex counting questions. The number of objects in each image ranges from 0 to 15. We randomly sampled 10 questions per ground truth count from the complex counting questions in the test set. This resulted in 149 complex counting questions in total, since some ground truth values have fewer than 10 samples available. We refer to this benchmark as the TallyQA-Complex benchmark.

We compare our method, using GPT-4o and Qwen2 VL 72B AWQ as the LVLM, against their corresponding base models. For reference, we also report the cross-domain performance of three of the best-performing trained counting models—GroundingREC, CountGD, and DAVE$_{prm}$—as well as the only prior training-free model, TFOC. The results are shown in Table 25. Note that CountGD and DAVE$_{prm}$ use weights trained on FSC-147, while GroundingREC uses weights trained on the REC-8K dataset (Dai et al., 2024), which its authors introduced alongside the model. Our method improves the MAE over the base LVLMs. Moreover, note that SOTA counting models are outperformed by the base LVLM models, further confirming that LVLMs possess better generalization and complex reasoning capabilities for counting tasks.

Table 25: Evaluation of the performance of LVLMs and their corresponding LVLM-Count as well as SOTA counting models on the TallyQA-Complex counting benchmark.

| Method | MAE ↓ | Δ | RMSE ↓ | Δ |
|---|---|---|---|---|
| TFOC (Shi et al., 2024) | 12.41 | - | 22.80 | - |
| DAVE$_{prm}$ (Pelhan et al., 2024) | 28.36 | - | 57.56 | - |
| CountGD (Amini-Naieni et al., 2024) | 9.78 | - | 17.21 | - |
| GroundingRec (Dai et al., 2024) | 5.83 | - | 10.13 | - |
| GPT4o | 2.60 | - | 4.74 | - |
| LVLM-Count (GPT4o as LVLM) | 2.28 | ↓ 0.32 | 4.18 | ↓ 0.56 |
| Qwen2 VL 72B AWQ | 3.21 | - | 5.35 | - |
| LVLM-Count (Qwen2 VL 72B AWQ as LVLM) | 2.47 | ↓ 0.74 | 4.35 | ↓ 1.00 |

