# OpenReview forum: "LVLM-Count: Enhancing the Counting Ability of Large Vision-Language Models"
_TMLR — Accepted by TMLR_

### Review · Reviewer_bvKd · 2025-10-03

**Summary Of Contributions:**

This paper identifies a real weakness in current LVLMs: they perform reasonably when only a few objects need to be counted, but their accuracy drops sharply as the numbers grow. To address this, the authors introduce LVLM-Count, a training-free, plug-and-play pipeline that combines expression extraction, area detection, segmentation, and—most notably—an object-aware division mechanism. The division avoids splitting through objects by combining clustering and A* path planning, which is both elegant and practical. The work also contributes a new benchmark, Emoji-Count, that stresses fine-grained distinctions and highlights where models still fail.

**Audience:**

Yes

**Audience Explanation:**

Likely high. A training-free, drop-in baseline that helps counting is useful to both researchers and engineers. The division idea may transfer to other dense-reasoning tasks.

**Broader Impact Concerns:**

Stronger counting helps ecology/field studies but raises privacy risks in surveillance. I’d encourage deployment guidance (schema checks, zero-object guards) and reporting error bars to avoid over-trusting counts in the wild.

**Claims And Evidence:**

Yes

**Claims Explanation:**

The paper evaluates multiple LVLMs across FSC-147, PASCAL VOC, Emoji-Count, and a Penguin benchmark with heavy occlusion. Across these, LVLM-Count consistently lowers MAE/RMSE vs. base models; e.g., on Penguin, GPT-4o MAE drops from 36.01 → 26.95 (main variant), with similar gains for Gemma 3 and Qwen2; a SAM-only variant also helps, underscoring robustness. The pipeline and object-aware division are described concretely (A* over mask graphs), and ablations/coarse analysis indicate why naïve splitting fails while object-aware paths help. The paper also reports the modest inference-time overhead (e.g., GPT-4o: 1.68s → 2.56s), and even notes that CoT prompting can hurt relying on VOC—usefully candid. Two caveats: statistics (variance/significance) are light, and some regressions exist on specific cases, but overall, the evidence is careful and convincing.

**Requested Changes:**

1. Add variability stats (95% CIs or bootstrapped SEs) for the headline numbers; call out regressions.
2. Describe Emoji-Count in more detail: categories, sampling strategy, splits, and licensing. Provide class definitions/splits and usage terms so others can adopt the benchmark.
3. Offer a clearer strategy for parsing outputs of open-source LVLMs (to handle non-numeric responses). Document the enforced schema (e.g., [[N]]) and failure rates; release the exact regex/parsers.
4. Quantify the “false positive counts” on zero-object images and suggest remedies. Quantify the non-zero predictions and propose a guardrail.
5. Gather implementation details (thresholds, NMS, erosion, mask refinements) into a single reproducibility table. Consider releasing a minimal config file.
6. The resource link in §4 returned “The requested file is not found” when I checked (Oct 3, 2025, PST). Please fix and ensure code/data are accessible.

---

### Review · Reviewer_ECMU · 2025-10-16

**Summary Of Contributions:**

**Contributions:**
The paper combines a pipeline of three types of VLMs (GroundingDINO, SAM, multimodal LLM such as Qwen2-VL) with a  more traditional computer vision algorithm for the task of counting.
To me the main contribution is enhancing existing “end-to-end” LLMs with more rule-based or structured steps that help with hard counting tasks. Just like in humans, once we go beyond a couple of items (as in this paper 20 or more items often), we need “external help” to keep track of things and categorize them: using our fingers to count, pointing individually at each item with our index finger to not double count, or going from top left to bottom right to not double count.
So it makes sense to not rely on an end-to-end pipeline.

The authors evaluate 3 off-the-shelf multimodal LLMs on 3 existing benchmarks as well as a new emoji-based counting tasks (an additional contribution).

They show that while VLMs are great on counting a few objects (e.g. below 20), they struggle with large numbers off-the-shelf. While their proposed LVM-Count method does not alleviate the issue completely, models are now closer to the true number.

**Good:**

The paper adds a straightforward well-defined contribution to the literature, for a task where it does makes sense to combine end-to-end neural models (LLMs) with structured models or rules.

The paper is easy to follow. Especially the sequence of figures make it easy to understand the method in Section 3.

Finally, from the appendix it is clear the authors have ran many additional experiments and explorations.

**Weaknesses:**

At the beginning of section 4.2 the authors list four different details that are in different appendix sections. Instead of just listing them, it would be good to summarize such findings in the main paper nonetheless and then refer to more details in the appendix. Reading so many references to appendix without any actual insights can break the reading flow.
Especially something like Appendix K, where you show how established non-LLM baselines perform, should definitely go in the main paper and main tables.

In general there is several of ablations or baselines that come up during reading the paper and are not addressed (see below).

**Additional Comments:**

It would be interesting to address how counting (at least when framed as this more hybrid approach with several steps of segmentation etc) is not just a trivial extension of the segmentation task? What are unique challenges beyond segmentation and maybe referring expressions? Related work (2nd paragraph) seems to address this a little but does not say how it falls short exactly.

I kept wondering as a reader: What exactly is the challenge here beyond established task formulations such as segmentation/referring expressions? And, is this pipeline the best way to address it?

**Audience:**

Yes

**Audience Explanation:**

multimodal LLMs are widely used and counting is one of their established failure modes. Thus, many researchers will care about how to address this.

**Broader Impact Concerns:**

The datasets used do not contain people, so I don't see direct ethical concerns. Fine-grained visual recognition can always be used in surveillance scenarios, but I do not think this specific paper requires a full discussion on this.

**Claims And Evidence:**

Yes

**Claims Explanation:**

The claim of the paper is simple “Here is a method to improve on top of existing LVLM’s counting ability”, and the experiments show this accurately.

**Requested Changes:**

I believe based on the definition of TMLR what counts as acceptance (“claims made in the submission supported by accurate, convincing and clear evidence”), this paper matches the TMLR criteria. But I do think some crucial things are still missing, open questions or ablations a reader might want answers for.

**Needed changes:**

Move more appendix ablations into the main paper and integrate them appropriately into the main flow. As is, there is very little analysis or ablations after showing the main results table.
Especially Appendix K contains very interesting baselines that often surpass the current method. This needs to be in the main paper, and contextualized: Why would one still prefer the given method? Why are those baselines good on some datasets and not others?

**Optional:**

1. Are there simpler simpler/different approaches?
So if the SAM model already segments the objects, why still ask LVLM to count? Can’t you just count the segmentation masks rule-based, making sure to not double-count?
Or ask the LVLM/CLIP etc to classify each segmented area individually?
E.g. the pipeline would be the same but in the step where you divide the image with the path finding algorithm, you instead pass each segmented area to a LVLM/CLIP?
This would result in a lot of LVLM calls per example, but would be worth to know as a reference?
Would this not work because as you said SAM/GroundingDINO are not perfect so passing larger chunks of images to the LVLMs mitigates that?
It would be great to explore why asking the LLM to count in the last step of the pipeline is needed.

2. Another ablation: The path finding algorithm vertically splits the image into 3-4 parts (based on examples I see in the appendix). Why not more splits, both vertical and horizontal, such that the LLM is asked to count on e.g. 5 or 10 sub-images?

3. Section “3.3 Object-aware division” is confusing to follow → e.g. clarify that choosing x-axis is arbitrary and you might as well choose y-axis? Or could you choose both? Please try to make this section less about explaining the intricacies of the algorithm and more about the high-level design choices. Perhaps the actual algorithm could just be references to the appendix and just said it is a path finding algorithm to split the image into parts, such that we don’t cut through images.

---

### Review · Reviewer_ZbBN · 2025-11-22

**Summary Of Contributions:**

The authors introduce a method to improve LVLMs' counting abilities. They main idea is to improve the division of images in a divide-and-conquer approach such that objects are not split apart. This improves the accuracy especially in images with many objects for several tested models (GPT-4o, Gemma 3, Qwen2). The idea of object-aware division is simple and straightforward, well explained, and experiments show that it is effective.

**Additional Comments:**

Given that this is not my primary area of expertise, I am not familiar with the state-of-the-art baselines the authors could have benchmarked against, so I cannot assess the completeness of their comparisons.

**Audience:**

Yes

**Audience Explanation:**

Counting objects in images is an important real-world task that state-of-the-art methods cannot handle sufficiently well, yet. The paper improves results of this task successfully.

**Broader Impact Concerns:**

Not applicable.

**Claims And Evidence:**

Yes

**Claims Explanation:**

Yes, the claims are supported by experiments performed on several datasets (FSC-147, PASCAL VOC Benchmark, Emoji Count, Penguin Benchmark). The MAE and RMSE of the counted number of objects is consistently improved compared to the baseline approaches. Visualizations in the appendix show some of the divisions and look reasonable.

**Requested Changes:**

The paper lacks some details to make results reproducible:

Q1) page 6: what is the "very low" value you set for GroundingDINO?

Q2) When using MeanShift Clustering, which version of it do you use (maybe I missed the citation)? Which bandwidth do you use? This can change the results significantly.

Q3) What is the runtime and complexity of your method?

Q4) On page 7 you state "for images with a very large number of objects, sometimes LVLMs refuse to count" - When or how often does this happen? Why does this happen? For which LVLMs does this happen? What changes if just a part of the image is given?

Q5) Can you elaborate more on the limitations? E.g., can the erosion cause very small objects to disappear? Experiments on images with very small objects that should be counted would be interesting.

---

> ### Author Response · Authors · 2025-11-26
> **Response to Reviewer ZbBN - Part I**
>
> We would like to extend our gratitude to you for reading our manuscript and giving us the opportunity to improve our work by providing valuable insights and comments. Before we present our response to each comment, we would like to mention that the revisions specific to your comments and concerns, have been highlighted in “purple" in the revised manuscript.
>
> The following are our responses and actions in regard to your comments:
>
> >Q1) page 6: what is the "very low" value you set for GroundingDINO?
>
>
> Thank you for your comment. A detection threshold of 0.1 is used for both the area detection and target segmentation stages. We report this value in the first paragraph that discuses the results for the FSC-147 dataset in Section 4.2 of the manuscript.
>
> The detection models propose bounding boxes for objects of interest and assign a score between 0 and 1, called the confidence score. The larger the score, the more confident the model is that it has detected an object of interest. The detection threshold, or confidence threshold, controls which bounding boxes are selected as the model's final output.
>
> For example, if a high value such as 0.9 is set as the detection threshold, the model only outputs detections it is very sure about. This increases precision, meaning the detections are very likely to be correct, but the user might miss some actual target objects that the model assigned a lower confidence score. Conversely, choosing a low value such as 0.1 increases recall, ensuring that the model does not miss actual targets, even if they were given a low confidence score.
>
> As mentioned in the paper, this might cause many false positive detections. However, our method is not sensitive to false positives since the counting is not based on the masks. The masks only determine which objects should not be cut by the division lines. The counting is done at the final stage by providing the subimages obtained from the object-aware division stage to an LVLM. Thus, false positives will simply be preserved from being cut, but they will be ignored by the LVLM when it comes to counting.
>
> >Q2) When using MeanShift Clustering, which version of it do you use (maybe I missed the citation)? Which bandwidth do you use? This can change the results significantly.
>
> Thank you for your comment. For the mean shift algorithm, we use the implementation from the Scikit-learn library [`sklearn.cluster.MeanShift()`]. This implementation is based on [1], with the additional feature that if the user does not specify a bandwidth, it is automatically estimated using `sklearn.cluster.estimate_bandwidth`. In other words, we do not provide any bandwidth to the function; rather, it is estimated by the module mentioned above without user intervention.
>
> In the revised manuscript (page 6, Section 3.3, second paragraph), we have added a reference to the original paper that proposed the mean shift algorithm and a footnote clarifying our use of the Scikit-learn implementation.
>
> > Q3) What is the runtime and complexity of your method?
>
> Thank you for this comment. In our manuscript, "Appendix J" discusses the inference time, including an analysis of the pipeline's individual stages and a comparison of the overall inference time with the base LVLMs. For your convenience, we present the tables from the Appendix J and a a summary of the discussion in that section here too.
>
> The table below shows the average inference time for each step in the pipeline on samples from the FSC-147 dataset.
>
> **Inference time breakdown of the LVLM-Count pipeline for different LVLMs (Time in seconds)**
> | Method | Extract $E$ from $Q$ | Area Detection | Target Segmentation | Clustering | $A^*$ | Subimage Analysis | Total |
> |-----------------------------|---------------------:|---------------:|--------------------:|-----------:|------:|------------------:|------:|
> | LVLM-Count (GPT4o) | 0.11 | 0.10 | 0.43 | 0.24 | 0.03 | 1.65 | 2.56 |
> | LVLM-Count (Gemma 3 27B) | 0.11 | 0.10 | 0.43 | 0.24 | 0.03 | 1.11 | 2.02 |
> | LVLM-Count (Qwen2 VL 72B AWQ)| 0.10 | 0.10 | 0.43 | 0.24 | 0.03 | 1.17 | 2.07 |
>
> The second table below shows the average total inference time on the FSC-147 samples for each base LVLM and the corresponding LVLM-Count. We can see that the inference time for LVLM-Count is greater than that of the base LVLM due to the additional stages in its pipeline. However, considering the improvement in counting accuracy, this increase in inference time might be acceptable for some applications.
>
> **Comparison of the inference time between the base LVLM and LVLM-Count**
> | Method | Inference (s) |
> |---------------------------------|---------------|
> | Base GPT4o | 1.68 |
> | LVLM-Count (Using GPT-4o) | 2.56 |
> | Base Gemma 3 27B | 1.52 |
> | LVLM-Count (Using Gemma 3 27B) | 2.01 |
> | Base Qwen2 VL 72B AWQ | 1.46 |
> | LVLM-Count (Using Qwen2 VL 72B AWQ) | 2.07 |

---

> ### Author Response · Authors · 2025-11-26
> **Response to Reviewer ZbBN - Part II**
>
> > Q4) On page 7 you state "for images with a very large number of objects, sometimes LVLMs refuse to count" - When or how often does this happen? Why does this happen? For which LVLMs does this happen? What changes if just a part of the image is given?
>
> Thank you for your comment. This phenomenon is common among all the LVLMs in our work. Based on our observations, when the total number of objects in the image exceeds 100, the LVLMs are more likely to exhibit this behavior. In these cases, the models avoid giving an explicit number and use general terms such as, "There are many objects-of-interest in the image." The average ground truth count for such cases in the FSC-147 dataset where GPT-4o prefers not to give an explicit number when prompted to count the objects-of-interest is 123.39. Nevertheless, please note that this issue is resolved by prompting the LVLM to estimate the number of objects-of-interest instead. Since our pipeline divides images into smaller parts, each containing fewer objects than the original image, the number of cases with this phenomenon was reduced by a factor of 2.19. The reason this phenomenon occurs is unclear to us, as it is inherent to these models and related to the inner dynamics of the learned weights.
>
> > Q5) Can you elaborate more on the limitations? E.g., can the erosion cause very small objects to disappear? Experiments on images with very small objects that should be counted would be interesting.
>
> Thank you for your comment. We use the erosion function solely to ensure that edge overlap between two masks does not obstruct the passage of a division line. For this reason, we have set the erosion kernel size to only 2 pixels. As a result, after applying the erosion kernel, a mask will disappear only if all its dimensions are less than two pixels. Objects in common datasets and scenes have dimensions much larger than this. In practice, a 2 pixel kernel only slightly erodes the edges of a mask. A visual example demonstrating that even very small masks are well preserved can be seen in "Figure 14, (a) and (b)". These images show the masks after erosion has been applied. By zooming in, it becomes clear that even very small masks are preserved and that the erosion does not cause any issues.
>
> ---
> Finally, we would like to express our gratitude once more for the time you spent reading our manuscript and our responses. We appreciate all of your comments, which constructively helped us to improve the quality of our manuscript. We hope that our efforts are to your satisfaction.
>
> ---
> References
>
> [1] Dorin Comaniciu and Peter Meer. Mean shift: A robust approach toward feature space analysis. IEEE Transactions on pattern analysis and machine intelligence, 24(5):603–619, 2002.

---

### Decision · Action_Editor_CEvh · 2026-01-30

**Recommendation:** Accept as is

**Audience:**

Yes

**Audience Explanation:**

While it is a specific application domain of VLMs, but it might be interesting for some of TMLR readers.

**Claims And Evidence:**

Yes

**Claims Explanation:**

The paper proposes a strong baseline for object count in LVLM using external models and object aware division algorithm. Rebuttal has addressed most of reviewers concerns and they agree that while the contribution is specific and incremental it stands as a good baseline for the object counting task with VLMs.